



# Wildfire smoke in the lower stratosphere identified by in situ CO observations

Joram J.D. Hooghiem[1], Maria Elena Popa[2], Thomas Röckmann[2], Jens-Uwe Grooß[3], Ines Tritscher[3], Rolf Müller[3], Rigel Kivi[4], and Huilin Chen[1]

[1]Centre for Isotope Research (CIO), Energy and Sustainability Institute Groningen (ESRIG), University of Groningen, Nijenborgh 6, 9747 AG Groningen, the Netherlands
[2]Institute for Marine and Atmospheric research Utrecht, Utrecht University, Princetonplein 5, 3508 TA Utrecht, the Netherlands
[3]Institute of Energy and Climate Research (IEK-7), FZ Jülich, Germany
[4]Space and Earth Observation Centre, Finnish Meteorological Institute (FMI), Tähteläntie 62, 99600 Sodankylä, Finland.

**Correspondence:** Huilin Chen (huilin.chen@rug.nl)

**Abstract.** Wildfires emit large quantities of aerosols and trace gases, which occasionally reach the lower stratosphere. In August 2017, several pyro-cumulonimbus events injected a large amount of smoke into the stratosphere, observed by lidar and satellites. Satellite observations are in general the main method of detecting these events since in situ aircraft- or balloon-based measurements of atmospheric composition at higher altitudes are not made frequent enough. This work presents accidental
balloon-borne trace gas observations of wildfire smoke in the lower stratosphere, identified by enhanced CO mole fractions at approximately 13.6 km. In addition to CO mole fractions, $CO_2$ mole fractions as well as isotopic composition of CO ($\delta^{13}$C and $\delta^{18}$O) have been measured in air samples collected using an AirCore and a LIghtweight Stratospheric Air sampler (LISA) flown on a weather balloon from Sodankylä (4–7 September 2017, 67.37° N, 26.63° E, 179 m.a.s.l), Finland. The greenhouse gas enhancement ratio ($\Delta$CO : $\Delta CO_2$) and the isotopic signature based on $\delta^{13}$C(CO) and $\delta^{18}$O(CO) independently identify
wildfire emissions as the source of the stratospheric CO enhancement. Back-trajectory analysis, performed with the Chemical Lagrangian Model of the Stratosphere (CLaMS), corrected for vertical displacement, due to heating of the wildfire aerosols, by observations made by the Cloud-Aerosol Lidar with Orthogonal Polarization (CALIOP) instrument, trace the smoke's origin to wildfires in British Colombia with an injection date of 12 August 2017. Knowledge of the age of the smoke allowed for a correction of the enhancement ratio, $\Delta$CO : $\Delta CO_2$, for the chemical removal of CO by OH. The stable isotope observations
were used to estimate the amount of tropospheric air in the plume at the time of observation to be about $34 \pm 14$ %. The in situ observations provide information on the trace gas chemistry of smoke plumes that reach the stratosphere, as well as the vertical extent of 1 km of the 2017 smoke plume.

## 1 Introduction

Wildfires emit a large quantity of polluting trace gases and aerosols into the atmosphere (Crutzen and Andreae, 1990; Andreae,
2019). These trace gases and aerosols affect the radiative transfer properties of the atmosphere and lead to the formation of tropospheric ozone. Not only the troposphere is affected, but the smoke also occasionally reaches the lower stratosphere





(Waibel et al., 1999; Fromm et al., 2000; Fromm and Servranckx, 2003; Jost et al., 2004; Fromm et al., 2010), enhancing aerosol levels and ozone (Fromm, 2005), with potential global effects (Peterson et al., 2018).

In 2017, a large smoke plume in the stratosphere was observed by ground-based LIDAR and the Cloud-Aerosol Lidar with Orthogonal Polarization (CALIOP) aboard the CALIPSO satellite (Khaykin et al., 2018). This smoke was attributed to Canadian forest fires, injected by pyro-cumulonimbus (pyro-Cb) events. The cumulative smoke mass injected into the stratosphere by five distinct pyro-Cb events was estimated to be 0.1 to 0.3 Tg (Peterson et al., 2018). The smoke mass density was further characterized using the Aerosol Robotic Network (AERONET) and Moderate Resolution Imaging Spectroradiometer (MODIS) (Ansmann et al., 2018), and the micro physical properties of the smoke were determined by LIDAR studies (Haarig et al., 2018; Hu et al., 2019; Baars et al., 2019).

Past injections of wildfire smoke into the stratosphere were mainly identified and characterized using satellite observations (e.g. Fromm et al., 2010). Nevertheless, wildfire smoke has been observed from in situ aircraft measurements as well. First, Waibel et al. (1999) reported a CO-plume in the Northern Hemisphere (NH) extra-tropical lowermost stratosphere at 10 km altitude. The plume was associated with the extensive 1994 burning season. In addition, Hudson et al. (2004), Ray et al. (2004), and Jost et al. (2004) found several smoke layers between 14.7–15.8 km ($\theta = 368$ to $393$ K). The enhanced levels of CO, up to 193 ppb, were found in the NH subtropical lower stratosphere (25° N), which was 1.3 km above the local tropopause. They attributed the origin of the smoke to North American forest fires. Finally, Cammas et al. (2009) reported on the injection of a smoke plume into the stratosphere also associated with North American forest fires.

In situ observations of wildfire smoke are typically identified by an increase in mole fractions of CO (Waibel et al., 1999; Jost et al., 2004; Cammas et al., 2009). In addition to CO, Cammas et al. (2009) measured $O_3$, $NO_x$, and PAN. These measurements correlate well with CO and are thus additional tracers for wildfire smoke. Furthermore, Hudson et al. (2004) and Jost et al. (2004) measured particle mass spectra containing carbon, potassium, organics, and ammonium ions. The stratospheric particle mass spectra were compared to mass spectra obtained from direct smoke measurements in the troposphere (Hudson et al., 2004), confirming the presence of smoke in the stratosphere.

Wildfire smoke has distinct trace gas source signatures. One way to identify the source of smoke is by using the enhancement ratio of $\Delta CO : \Delta CO_2$ (Mauzerall et al., 1998), and another way is to use the stable isotopic composition of CO (Brenninkmeijer et al., 1999; Kato et al., 1999b; Röckmann et al., 2002). Only Jost et al. (2004) measured $CO_2$, allowing $\Delta CO : \Delta CO_2$ to be quantified, confirming the smoke's origin. These source signatures have been successfully used in many ground-based and airborne studies on wildfire smoke plumes in the troposphere (e.g. Andreae et al., 2001; Bergamaschi et al., 1998; Tarasova et al., 2007).

This work presents the first balloon-borne CO and $CO_2$ observation of a wildfire smoke plume in the stratosphere. The Air-Core sampling technique (Karion et al., 2010; Membrive et al., 2017) provides an accurate measurement of enhancement ratios of $\Delta CO : \Delta CO_2$; where the LIghtweight Stratospheric Air Sampler LISA (Hooghiem et al., 2018) is used to collect larger samples that allow for the determination of the carbon and oxygen stable isotopic compositions of CO. A back-trajectory analysis was performed using the Chemistry Lagrangian Model of the Stratosphere, CLaMS (McKenna et al., 2002), to determine the source region and fire date, which helps in the quantification of the chemical loss of CO by OH. The trace gas and isotopic





composition measurements are used to confirm the wildfire smoke burning origin. Finally the observations are used to estimate the fraction of tropospheric air in the enhanced smoke plume.

## 2 Methods

### 2.1 Sampling instruments and flights

The air sampling was done with LISA (Hooghiem et al., 2018) and an AirCore (Karion et al., 2010). Both instruments are capable of sampling the stratosphere, and can be flown using small weather balloons which are easy to operate. We refer to the original references for the details; here we present a brief description.

An AirCore is a long coiled thin tube, with one end open and one end closed. The AirCore passively takes an air sample during descent, relying on increasing atmospheric air pressure to push air into the tube. A magnesium perchlorate dryer is positioned at the inlet in order to dry the incoming sample. The AirCore used in this study consisted of two pieces of stainless-steel tubing with SilcoNert 1000 coating (Restec Inc.) to create a chemically inert and smooth surface. The first section was a 40 m long tube with a 0.635 cm (1/4 inch) outer diameter; the second piece was a 60 m long tube, with 0.3175 cm (1/8 inch) outer diameter. Both tubes had a wall thickness of 0.0254 cm (0.01 inch). The benefit of an AirCore, when launched on a balloon, is the retrieval of an atmospheric profile. The amount of sample retrieved per flight, was in total 1.4 l at standard temperature and pressure, and roughly 0.3 l is stratospheric. The amount of sample is thus is limited.

Contrary to the AirCore, LISA takes four samples actively using a small pump upstream of four sampling bags. The active sampling results in a larger amount of sample, thus allowing for isotope analysis. The four sampling bags are filled at a different altitude between 12–25 km during ascent (Hooghiem et al., 2018). The ascent speed is usually slower than that during descent. Sampling during ascent thus favours a higher vertical resolution, which is around 0.5 to 1 km for the LISA samples.

Both LISA and the AirCore were launched together on the same balloon from the radiosonde facility of the Finnish Meteorological Institute at Sodankylä (67.37° N, 26.63° E, 179 m above mean sea level (a.m.s.l.)) using Totex TX3000 balloons. The balloons typically reach 30 km thus penetrating most of the atmospheric mass ($> 99$ %). Three flights with both instruments were performed, one on each day from 4 to 6 September 2017. Furthermore, the AirCore was flown without LISA on 7 September. Additional to the AirCore and LISA, a Vaisala radiosonde (RS-92SGP) was added to the payload for collocated temperature, pressure, and relative humidity measurements. Furthermore, the radiosonde collects GPS and altitude information during flight (Dirksen et al., 2014).

### 2.2 Measurements

#### 2.2.1 Mole fraction measurements of $CO_2$ and CO

Directly after the payload was retrieved from the landing location, the AirCore and LISA samples were analysed for $CO_2$, $CH_4$, and CO mole fractions using the Cavity Ring-Down Spectroscopy (CRDS) technique (analyser model: Picarro G2401 (AirCore), Picarro G2401-m (LISA) see e.g. (Crosson, 2008) for more information on the CRDS-analyser). A calibration gas


was also measured to link the mole fraction measurements to the following World Meteorological Organization, or WMO, scales: X2007 ($CO_2$) (Zhao and Tans, 2006), X2004($CH_4$) (Dlugokencky, 2005) and X2014A (CO). Throughout this work, the abbreviation ppm is used for $\mu mol\,mol^{-1}$, and ppb for $nmol\,mol^{-1}$.

After post-processing of the analyser output, dry mole fractions were obtained using the instrument specific but well-defined analyser response to $H_2O$ (Rella et al., 2013; Chen et al., 2010). This is especially relevant for the LISA samples (Hooghiem et al., 2018); the range of water vapour mole fractions was 0.03–0.15 %, partly because of diffusion into the sampling bag (Hooghiem et al., 2018). The mole fraction results of the AirCore were further processed to give vertical profiles as described in Karion et al. (2010); Membrive et al. (2017). The uncertainty of the AirCore measurements typically is 0.1 ppm for $CO_2$, 2 ppb for $CH_4$ and 2 ppb for CO. Measurements on samples collected with the LISA sampler have an uncertainty of 0.14 ppm for $CO_2$, 2.3 ppb for $CH_4$ and 7.8 pbb for CO. This uncertainty includes analyser precision, calibration transfer, a dead volume bias, and storage bias; for the technical details we refer to Hooghiem et al. (2018).

### 2.2.2 LISA sample transfer and storage

As described in Hooghiem et al. (2018), the bags used in the LISA sampler provide limited stability to the sample. Therefore, the samples were transferred into 350 ml glass flasks, after the CRDS analysis. The flasks have a Rotulex connection and are sealed with Viton-70 O-rings. The flasks and transfer lines were evacuated using a vacuum pump (flasks were evacuated using an Adixen Drytel 1025, the transfer lines using a Vacuubrand MD 1) before the samples were introduced into the flasks. As the bags are compressible, the sample was pushed into the flask until local ambient atmospheric pressure was reached, typically 950 hPa, or until the sample was fully expanded into the glass container at a pressure lower than ambient pressure. The air samples were stored in these glass flasks until they were analysed in the laboratory for the isotopic compositions of CO. The storage period was 3–7 months.

### 2.2.3 Analysis of stable isotopic composition of CO

The LISA samples were shipped to the Institute for Marine and Atmospheric research Utrecht (IMAU) for analysis of $\delta^{13}C$ and $\delta^{18}O$ in CO. The samples were analysed using Continuous-Flow Isotope-Ratio Mass Spectrometry (CF-IRMS) (Pathirana et al., 2015). The $\delta$-values for carbon are reported on the Vienna Peedee Belemnite scale (VPDB), whereas oxygen values are reported on the Vienna Standard Mean Ocean Water scale (VSMOW). For details about the measurements we refer the reader to Pathirana et al. (2015). Briefly, the sample is carried, using He as a carrier gas, through an Ascarite and magnesium perchlorate trap, removing $CO_2$ and $H_2O$ from the sample. $N_2O$ and any remaining $CO_2$ are removed by means of a cryogenic trapping using liquid $N_2$. Then the CO is converted to $CO_2$ with the aid of the Schütze reagent. A second cryogenic trap isolates the $CO_2$ derived from CO, which, after passing through a GC column, is fed to an IRMS via an open split system. The IRMS analyses $\delta^{13}C$ and $\delta^{18}O$. The $\delta^{18}O$ is corrected for the additional oxygen atom added in the conversion to $CO_2$ as described by (Pathirana et al., 2015). The CF-IRMS system in this study was the same as that described in (Pathirana et al., 2015), with one exception. The CF-IRMS analysis used about 150 ml sample, and requires a sufficiently high upstream pressure (> 900 hPa absolute) to maintain sample flow. As mentioned in Section 2.2.2, the starting pressure of the LISA samples was equal to or





lower than 950 hPa, and it decreased rapidly during a measurement due to the small flask volume of 350 ml. Therefore, the pressure in the flasks was increased during the measurement using CO-free synthetic air. Each sample was measured at least twice while being diluted in this way.

As the samples were measured at very low concentrations, meaning very low peak areas in the IRMS measurement, special
attention was paid to estimate the effect of non-linearity on the reported isotopic composition. Dilution tests showed detectable non-linear behaviour for $\delta^{18}O$ and $\delta^{13}C$ below a peak area corresponding to approximately 10 ppb and 15 ppb respectively, but none of the samples presented here were measured at such low peak areas. Therefore we can consider the non-linearity effect negligible.

The average analytical precision for this dataset, estimated from the reproducibility of repeated sample measurements, was
0.5 ‰ for $\delta^{13}C$ and 0.5 ‰ for $\delta^{18}O$.

## 2.3 Characterisation of the plume

### 2.3.1 Back-trajectory analysis

To determine the origin of the observed air masses with enhanced CO and $CO_2$ mole fractions (see Section 3.1), back-trajectories were calculated with the trajectory module of the Chemical Lagrangian Model of the Stratosphere (CLaMS)
(McKenna et al., 2002) driven by ERA-Interim meteorological data (Dee et al., 2011). The mixing (not used here) and advection schemes of CLaMS are capable of resolving 2-D filamentary structures that exist in the stratified stratosphere. The CLaMS trajectories are calculated on isentropic surfaces, with the vertical displacement from the isentropic surfaces calculated from diabatic vertical velocities (Ploeger et al., 2010).

Khaykin et al. (2018) showed the increased vertical transport of the wildfire smoke due to heating induced by aerosols. As
this additional vertical velocity component is not included in the calculation it is difficult to directly backtrack the air masses by a single trajectory. Therefore, correspondence with CALIOP elastic backscatter-ratio at 532 nm (Winker et al., 2010) is used to correct the back trajectories in a piecewise manner. A first set of trajectories was started backwards in time from the altitude of the observed CO-peak maximum ($p = 155$ hPa, 13.6 km) and from the altitudes where the CO-enhancement was half of the maximum (166 and 148 hPa, 13.3–13.8 km). After that, matches with CALIOP night-time observations were determined
as follows. For each orbit time, the minimum distance of the trajectory and the orbit location was calculated. Wherever the distance was below 250 km, the observed backscatter ratio was investigated for nearby un-typical aerosol enhancements. In this way the smoke could be accurately traced back to the injection date and region.





### 2.3.2 Determination of the source signature

The method usually employed to determine the source signature of a pollution source is to assume that a measured air parcel is
the result of mixing of background air and a polluting source, i.e. two end-member mixing. Then the following mass balance
applies to the mole fractions $x$:

$$x_{\mathrm{ap}} = f_{\mathrm{bg}} x_{\mathrm{bg}} + f_{\mathrm{src}} x_{\mathrm{src}} \tag{1}$$

where ap denotes the air parcel, bg means background, and src means the pollution source. A similar balance can be written
for the stable isotopes.

$$x_{\mathrm{ap}} \delta^{13}\mathrm{C}_{\mathrm{ap}} = f_{\mathrm{bg}} x_{\mathrm{bg}} \delta^{13}\mathrm{C}_{\mathrm{bg}} + f_{\mathrm{src}} x_{\mathrm{src}} \delta^{13}\mathrm{C}_{\mathrm{src}} \tag{2}$$

Equation 1 and Equation 2 can be solved to yield:

$$\delta^{13}\mathrm{C}_{\mathrm{ap}} = \left( \delta^{13}\mathrm{C}_{\mathrm{bg}} - \delta^{13}\mathrm{C}_{\mathrm{src}} \right) \frac{f_{\mathrm{bg}} x_{\mathrm{bg}}}{x_{\mathrm{ap}}} + \delta^{13}\mathrm{C}_{\mathrm{src}} \tag{3}$$

Which results in a linear relation between $\delta^{13}\mathrm{C}_{\mathrm{ap}}$ and $x_{\mathrm{ap}}^{-1}$ if $\left( \delta^{13}\mathrm{C}_{\mathrm{bg}} - \delta^{13}\mathrm{C}_{\mathrm{src}} \right) f_{\mathrm{bg}} x_{\mathrm{bg}}$ is assumed to be constant. This relation
was first recognized by Keeling (1958) and is a special case of the more general Miller-Tans method (Miller and Tans, 2003).
A relation for $^{18}\mathrm{O}$, equivalent to Equation 3, can be derived as well.

This relation can be safely applied to CO if it can be assumed that removal by OH is negligibly small. However, the age of
the observed air parcel, 24–25 days (see Section 3.2), is too old to ignore the chemical reaction of $\mathrm{CO + OH}$. Therefore the
evolution of CO in the plume is modelled as follows:

$$\frac{dn(\mathrm{CO})}{dt} = -k_{\mathrm{er}} \left( n(\mathrm{CO}) - n_{\mathrm{bg}}(\mathrm{CO}) \right) - k_1 n(\mathrm{OH}) n(\mathrm{CO}) \tag{4}$$

where $k_{\mathrm{er}}$ is the entrainment rate in $\mathrm{s}^{-1}$. $n(\mathrm{X})$ is the number density of species X, in $\mathrm{cm}^{-3}$. The reaction rate of the reaction
$\mathrm{CO + OH}$, $k_1$, was taken from McCabe et al. (2001):

$$k_1 = 1.57 \cdot 10^{-13} + 3.54 \cdot 10^{-33} n \tag{5}$$

in $\mathrm{cm}^3 \ \mathrm{s}^{-1}$, where $n$ is the number density of air in $\mathrm{cm}^{-3}$. The number density of OH is taken to be $2.7 \cdot 10^6 \ \mathrm{cm}^{-3}$ between
0–4 km altitude, $1.6 \cdot 10^6 \ \mathrm{cm}^{-3}$ between 4–8 km altitude and $1.2 \cdot 10^6 \ \mathrm{cm}^{-3}$ for altitudes $>8$ km (Mauzerall et al., 1998),
although Mauzerall et al. (1998) specifies this value for the range 8–12 km. The production of CO from NMHCs and $\mathrm{CH}_4$ is
assumed to be insignificant, and therefore neglected.

After separating the variables of Equation 4, integrating, and solving for $n(\mathrm{CO})(t)$ using the boundary condition that at
$n(\mathrm{CO})(t=0) = n_0(\mathrm{CO})$, this yields (see Section A for the derivation):

$$n(\mathrm{CO})(t) = \left( n_0(\mathrm{CO}) + \frac{k_{\mathrm{er}} n_{\mathrm{bg}}(\mathrm{CO})}{-k_{\mathrm{er}} - k_1 n(\mathrm{OH})} \right) \exp\left( -\left( k_{\mathrm{er}} + k_1 n(\mathrm{OH}) \right) t \right) - \frac{k_{\mathrm{er}} n_{\mathrm{bg}}(\mathrm{CO})}{-k_{\mathrm{er}} - k_1 n(\mathrm{OH})} \tag{6}$$





**Table 1.** The coefficients $a$, $b$, and $c$ for Equation 7 for $^{13}C$ and $^{18}O$.

| Isotope | $a$ | $b$ | $c$ |
|---|---|---|---|
| $^{13}C$ | -0.00655 | 0.02269 | 0.00947 |
| $^{18}O$ | -0.01191 | 0.00603 | -0.00341 |

**Table 2.** Four main sources of CO, and their isotopic source signatures. [a](Stevens et al., 1972), [b](Brenninkmeijer, 1993), [c](Stevens and Wagner, 1989), [d](Bergamaschi et al., 1998), [e](Saurer et al., 2009), [f](Manning et al., 1997), [g](Brenninkmeijer and Röckmann, 1997), and [h](Vimont et al., 2019). Table based on the most recent compilation of source signatures by Vimont et al. (2019).

| Source | $^{13}C$ (VPDB) | Uncertainty | $^{18}O$ (VSMOW) | Uncertainty |
|---|---|---|---|---|
| Fossil fuel combustion[a,b] | $-27.5\,‰$ | $\leq 1\,‰$ | $23.5\,‰$ | $\leq 1\,‰$ |
| Biomass burning[c,d,e,f] | $-12 - -25\,‰$ | $1–3\,‰$ | $10–18\,‰$ | $1–3\,‰$ |
| $CH_4$ oxidation[f,g] | $-52.6\,‰$ | $1–3\,‰$ | $0\,‰$ | $> 3\,‰$ |
| NMHC oxidation[c,g,h] | $-32\,‰$ | $1–3\,‰$ | $0–4\,‰$ | $> 3\,‰$ |

An equivalent equation to Equation 6 can be written for $^{13}CO$ and $C^{18}O$. The reaction rates for these minor isotopologues can be determined from fractionation factors:

$$\alpha = \frac{k_{\mathrm{minor}}}{k_{\mathrm{major}}} = \frac{1}{a + bp + cp^2} \qquad (7)$$

where $p$ is the atmospheric pressure in bar. The coefficients $a$, $b$, and $c$ are fit to combined datasets from Röckmann et al. (1998); Stevens et al. (1980); Smit et al. (1982) and are obtained from Gromov (2013), and can be found in Table 1. $k_{\mathrm{minor}}$ and

$k_{\mathrm{major}}$ are the reaction rates of the rare and abundant isotopologues respectively. We assume that $k_{\mathrm{major}} = k_1$ as in Equation 5.

     Now $k_{\mathrm{er}}$ can be written as (see Section A for the derivation):

$$k_{\mathrm{er}} = \frac{-\ln(1 - f_{\mathrm{strat}})}{t} \qquad (8)$$

If the age, $t$, of the air parcel is known, then $k_{er}$ can be evaluated as a function of $f_{\mathrm{strat}}$ which is the total fraction of stratospheric air entrained in the air parcel.

In principle $f_{\mathrm{strat}}$ is unknown, but it can be evaluated over the full range from 0 to 1 to give an idea what range of source signatures is. Then the source signature can be estimated, and compared to the known sources and their signatures, see Table 2. Furthermore, a best estimate can be made using $f_{\mathrm{strat}}$ found from Section 2.3.3.

     Temperature and pressure are known from the observation made by LISA; temperature and pressure are assumed to be constant. Then Equation 4 can be used to obtain the CO, $^{13}CO$, and $C^{18}O$ number densities from which $\delta^{13}C$ and $\delta^{18}O$ can

be determined at an earlier time compared to the observation.



### 2.3.3 Stratosphere-troposphere exchange estimate based on the in situ observations

By combining the CO isotope and mole fraction measurements in a simple mixing model, the contribution of tropospheric and stratospheric air to the observed plume can be quantified using 3 end-member mixing. Assuming mixing of stratospheric air, tropospheric air, and wildfire smoke, the mole fraction of the air parcel, called ap (air parcel), sampled by LISA is as follows:


$$x_{ap} = f_w x_w + f_t x_t + f_s x_s \qquad (9)$$

and for the stable isotopes:

$$x_{ap}\delta^{13}C_o = f_w x_w \delta^{13}C_w + f_t x_t \delta^{13}C_t + f_s x_s \delta^{13}C_s \qquad (10)$$

and

$$x_{ap}\delta^{18}O_{ap} = f_w x_w \delta^{18}O_w + f_t x_t \delta^{18}O_t + f_s x_s \delta^{18}O_s \qquad (11)$$

Here $f$ and $x$ denote volumetric air fraction and mole fraction respectively. Subscripts w, t, and s denote the "wildfire smoke", "tropospheric", and "stratospheric" end-members respectively, whereas ap denotes "air parcel". An end-member is defined here by its carbon and oxygen isotopic composition, and by its mole fraction, and it is possible to distinguish the air parcel from others based on those parameters. Mass balance requires:

$$f_w + f_t + f_s = 1 \qquad (12)$$

This is an extension of two end-member mixing presented by Equation 1 and Equation 2. Combining Equation 9, Equation 12, Equation 10, and Equation 11 yields a system of four linear equations:

$$
\begin{bmatrix}
1 & 1 & 1 \\
x_w & x_t & x_s \\
x_w\delta^{13}c_w & x_t\delta^{13}C_t & x_s\delta^{13}C_s \\
x_w\delta^{18}O_w & x_t\delta^{18}O_t & x_s\delta^{18}O_s
\end{bmatrix}
\begin{bmatrix}
f_w \\ f_t \\ f_s
\end{bmatrix}
=
\begin{bmatrix}
1 \\
x_{ap} \\
x_{ap}\delta^{13}C_{ap} \\
x_{ap}\delta^{18}O_{ap}
\end{bmatrix}
+
\begin{bmatrix}
M_r \\
x_r \\
x_r\delta^{13}C_r \\
x_r\delta^{18}O_r
\end{bmatrix}
\qquad (13)
$$

The second term on the right-hand side, $[M_r, x_r, x_r\delta^{13}C_r, x_r\delta^{18}O_r]$, is the residual vector that ensures equality of the over-constrained problem, with $M_r$ denoting the normalized mass. Note that ideally this term is zero.

This set of equations is normalized to the observations and weighted so that all residuals, are of equal importance, except for the mass balance. The mass balance was given extra weight, which conforms to the assumption that the observed air parcel is purely a mixture of tropospheric, stratospheric, and wildfire smoke.

Equation 13 was solved for $f_w$, $f_t$, and $f_s$ by minimizing the residuals $[M_r, c_r, c_r\delta^{13}C_r, c_r\delta^{18}O_r]$ using a non-negative least square algorithm (Lawson and Hanson, 1995). The end-member definitions used are presented in Table 3. The stratospheric
end-member definition and the plume observation are based on the balloon-borne observations presented in this paper, and are well defined. The mole fractions are assigned an uncertainty equal to the measurement uncertainty, which, considering the





**Table 3.** End-member definitions and air parcel as used in the Monte-Carlo simulation. Mole fractions are given in ppb, and $\delta$-values in ‰. The large range for the plume mole fraction does not significantly affect the results. Here ap means air parcel, and is based on the in situ enhancement observation made by LISA, see Table 4. An additional Monte-Carlo run is made with the ap OH-Corrected. A number density OH of $1.2 \cdot 10^6$ cm$^{-3}$ is used. The stratospheric end-member definition is based on the in situ observation of stratospheric background presented in Table 4. The tropospheric and wildfire smoke end-member definitions are based on observations from earlier work, see text.

| Airmass | Variable | Mean | 1-$\sigma$ | Distribution |
|---|---|---|---|---|
| Wildfire smoke | CO | $0.5 \cdot 10^6$ to $1.5 \cdot 10^6$ | - | uniform |
| | $\delta^{13}$C | $-24.4$ to $-21.3$ | - | uniform |
| | $\delta^{18}$O | $16.3$ to $18.0$ | - | uniform |
| Stratosphere | CO | 34 | 8 | normal |
| | $\delta^{13}$C | $-1.0$ | 1.0 | normal |
| | $\delta^{18}$O | $-29.6$ | 1.0 | normal |
| Troposphere | CO | 72 | 8 | normal |
| | $\delta^{13}$C | $-32$ to $-28$ | - | uniform |
| | $\delta^{18}$O | $-4$ to $0$ | - | uniform |
| ap | CO | 74 | 8 | normal |
| | $\delta^{13}$C | $-28.8$ | 1.0 | normal |
| | $\delta^{18}$O | 4.3 | 1.0 | normal |
| ap OH-Corrected | CO | 111.0 | 8 | normal |
| | $\delta^{13}$C | $-26.2$ | 1.0 | normal |
| | $\delta^{18}$O | 9.2 | 1.0 | normal |

observations from AirCore, is also a reasonable measure of natural variability. Yet, for the stable isotopes a larger uncertainty was assumed, twice the measurement uncertainty, to account for unknown variability of the stratospheric background and variability within the plume.

On the other hand, the wildfire smoke and tropospheric end-member rely on measurements from earlier publications, which introduces a large uncertainty on end-member definitions. First of all, the isotopic composition and mole fraction of tropospheric CO exhibits temporal, both seasonal and annual, and latitudinal gradients (Bergamaschi et al., 2001; Mak et al., 2003), which complicates the end-member definition of the tropospheric end-member. The smoke source region lies between 65°– 75° N (see Section 3.2), with south-westerly surface winds coming from the Pacific Ocean (Peterson et al., 2018). Tropospheric

CO mole fractions used are based on measurements of CO from Midway island (Petron et al., 2019), and reported to be typically $72 \pm 8$ ppb in the six weeks preceding the event. The range of isotopic-composition values considered are therefore obtained from measurements made at Izana (28° N and 16° W, which is representative for CO in air that travels over the ocean at mid-latitudes. $\delta^{13}$C ranges between $-32$ and $-28$ ‰ and $\delta^{18}$O ranges between $-4$ and $0$ ‰ (Bergamaschi et al., 2001; Mak





et al., 2003). The mole fractions obtained here from Midway island are consistent with those co-reported with the $\delta^{13}$C and

$\delta^{18}$O ranges by Bergamaschi et al. (2001); Mak et al. (2003).

In addition, the wildfire smoke signature, see Table 2, is subjected to a large variability due to the type of burned plants (categorised as C3 or C4, with a difference in photosynthesis), the burning temperature, and possibly the groundwater isotopic composition (Kato et al., 1999a). Since the wildfire originated in Canada, the fuel consisted mainly of C3 plants which are typically more depleted in $\delta^{13}$C. Furthermore, the fire was energetic enough to trigger a pyro-Cb event, which makes it

reasonable to assume that it was in an efficient burning regime, which typically leads to higher $\delta^{18}$O. Therefore, the range of isotopic composition of CO of the wildfire smoke assumed here is according to atmospheric measurements around forest fires (Brenninkmeijer et al., 1999), see Table 3. This thus differs from the isotopic composition displayed in Table 2.

The mole fraction of wildfire smoke is also an unknown, but is however much larger than both the stratospheric and tropospheric background mole fraction. In fact, a large mole fraction ensures, by virtue of Equation 12, that $f_{\rm w} \ll f_{\rm s}$ and $f_{\rm w} \ll f_{\rm t}$.

In order to capture the above-mentioned uncertainty, a Monte-Carlo simulation was performed. The variable input parameters are randomly drawn from the respective distributions assumed, also presented in Table 3.

Since the mole fraction and isotopic composition of the smoke plume and the tropospheric end-members are ill-defined, partially due to natural variability, the Monte-Carlo results are filtered. Solutions to Equation 13 are only allowed if the residuals are smaller than the measurement 1-sigma uncertainty attributed to our stratospheric observations, e.g. 4 ppb for the mole

fractions and 1 ‰ for the $\delta$-values. A solution is thus only allowed when its consistent with the observations and the reported isotopic composition range published in literature. Furthermore, the solution requires all fractions to be larger than 0, where four significant figures were considered, to avoid a large amount of unrealistic solutions.

## 3   Results

### 3.1   Observation of a stratospheric CO enhancement

The CO measurements of both the LISA sampler and the AirCore are presented in Figure 1a. A clear carbon monoxide enhancement was observed between 13 and 14 km altitude on 4 and 5 September 2017. The plume was well above the tropopause which can be seen from the CO gradient below 13 km. The tropopause height was determined based on the lapse rate from the radiosonde temperature measurements to be 12 km on both flights, confirming that the observed plume is above the tropopause. The potential temperature was $\theta \approx 350$ K at 13 km and $\theta \approx 380$ K at 14 km, which classifies this part of

the stratosphere as the extra-tropical lowermost stratosphere (Holton et al., 1995). The observed $CO_2$ mole fraction Figure 1b showed a slight increase in the same layer, which allowed for determination of the enhancement ratio, $\Delta CO : \Delta CO_2$ see Section 3.3.





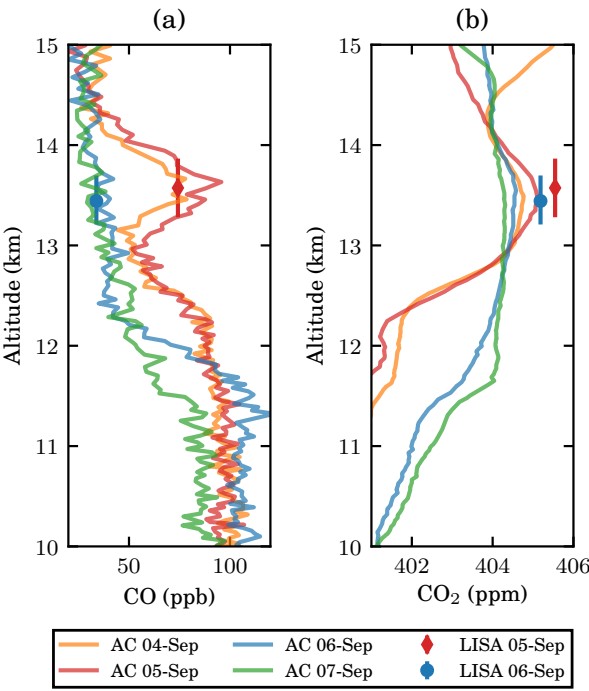

**Figure 1.** The CO, (a), and $CO_2$, (b), profiles from AirCore (lines, abbreviated as AC in the legend) and the LISA sampler (markers). The profiles are shown between 12 and 15 km altitude and are coloured by date. The LISA sampler vertical error bars represent the total vertical coverage of the sample, with the mean altitude as shown. For mole fraction measurement uncertainties, the reader is referred back to Section 2.2.1.

## 3.2 Origin and age of the plume based on back-trajectory analysis

Figure 2a shows the CALIOP backscatter ratio at 532 nm ($R_{532}$) as well as the location the of back-trajectory result on

3 September, with a match distance of 47 km. The match distance of the trajectories from the centre and upper altitude are above 250 km (282 and 434 km) and are not shown here. To track the aerosol cloud, new back-trajectories were initialized starting from 3 September exactly where aerosol cloud is observed in the CALIOP data (Figure 2b white crosses). Again matches with CALIOP orbits were calculated. Figure 2c and d are similar to Figure 2a and b but for 20 August, where a difference in altitude is observed. This altitude mismatch might be due to the additional vertical motion related to additional

radiative heating of the smoke plume, which is absent in the CLaMS model. Similarly to 3 September, in a third step, back-trajectories were calculated from this observed aerosol cloud. On 14 August, the back tracked air parcels are within the range of the aerosol cloud that was observed by the Ozone Mapping Profiler Suite (OMPS), as shown by Peterson et al. (2018). The result of the back-trajectory analysis is shown to match the location of the aerosol enhancement observed by OMPS and CALIOP in Figure 3.

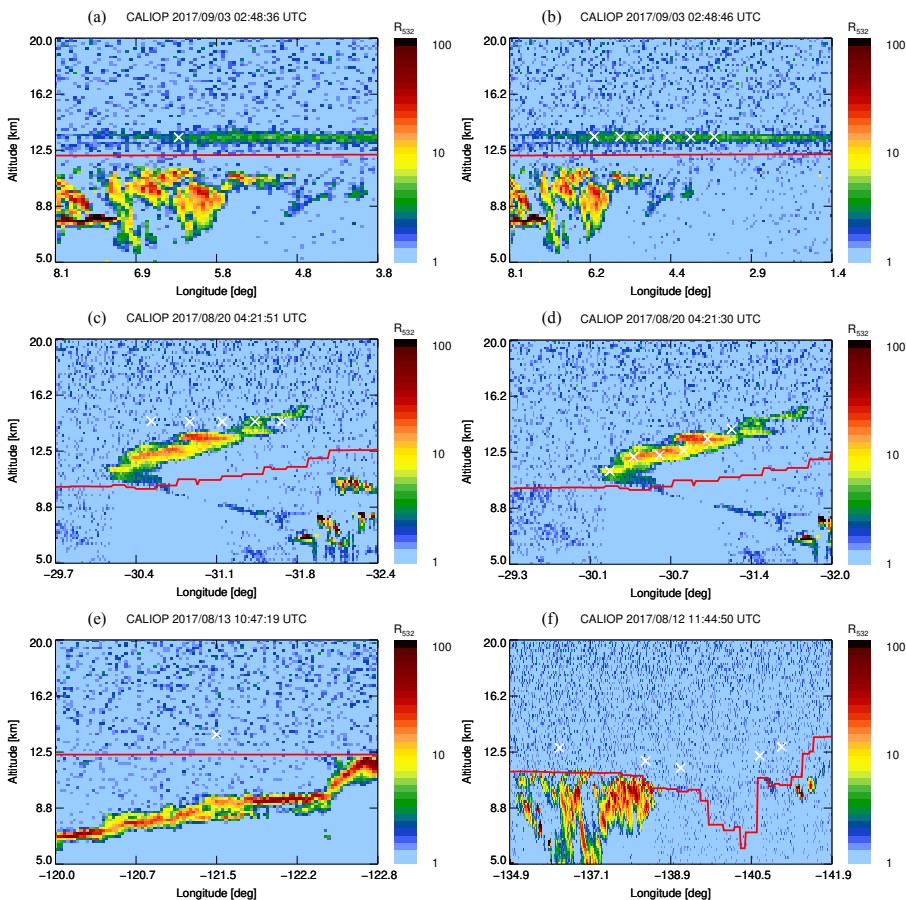

**Figure 2.** Altitude-Longitude plot with CALIOP backscatter ratio and back-trajectory results. The red line indicates the thermal tropopause from CALIOP. a) a match on 3 September between CALIOP and CLaMS back-trajectories starting on 5 September, the white cross. The distance between the back-trajectory result and the plotted white cross is 47 km. b) newly initialized back-trajectories on 3 September, white crosses. c) the matches on 20 August, between CALIOP backscatter and the results from the back-trajectories initialized on 3 September, where the distance between back-trajectory result and CALIOP overpass is smaller than 250 km. Note the altitude discrepancy. During the period between 20 August and 3 September the correspondence was always sufficiently good. d) newly initialized back-trajectories on 20 August. e) Match location on 13 August with a vertical distance of 3 km to the aerosol plume. f) Match location on 12 August with no clear correspondence between the trajectory and the enhanced backscatter from CALIOP, the time of the location match is well before the fires, 21:00 UTC (Peterson et al., 2018).





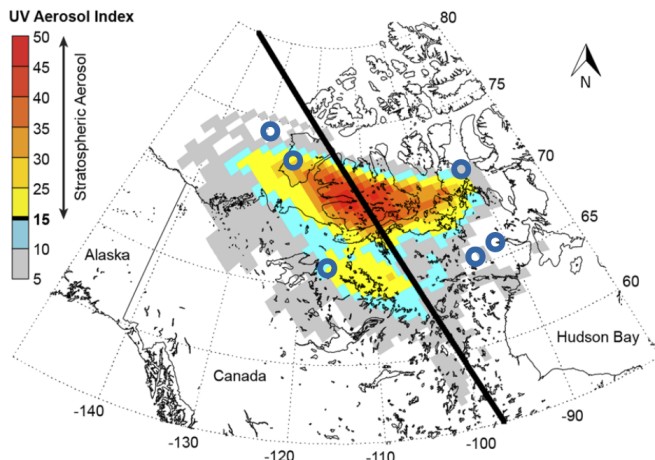

**Figure 3.** Figure from (Peterson et al., 2018, Fig. 3) showing the ultra-violet aerosol index from the Ozone Mapping Profile Suite (OMPS) on 14 August with the CALIPSO satellite track in black. Here the result of the back-trajectory on 14 August are added to the figure, blue circles, coinciding with the stratospheric smoke plume. The original figure was published under a Creative Commons Attribution 4.0 International License, http://creativecommons.org/licenses/by/4.0/.

Figure 2f shows the location on 12 August, before the injection. This cloud is located over British Columbia and caused by the pyro-convection as concluded by Peterson et al. (2018). Further back in time, the CALIOP backscatter data do not show any aerosol enhancement where a location match between CALIOP and the back-trajectory was found. Therefore, the origin of the observed plume could be confirmed by this piecewise trajectory analysis that accounts for the vertical transport due to heating caused by the fire. Furthermore, the age of the plume at the time of observation was 24–25 days.

**3.3   Enhancement ratios of CO/CO$_2$ of the plume**

Figure 4 shows a scatter plot of the observed CO and CO$_2$ mole fractions in the plume, both measured CO and CO$_2$ data and CO corrected for removal by OH (see below). The enhancement ratio on 4 September, $40 \pm 2 \, \mathrm{ppb \, ppm^{-1}}$, is higher than that on 5 September, $34 \pm 1 \, \mathrm{ppb \, ppm^{-1}}$. The mole fraction enhancement ratio, $34$–$40 \, \mathrm{ppb \, ppm^{-1}}$, falls in the range of fresh to aged biomass burning plumes, as defined by Mauzerall et al. (1998).

The chemical lifetime of CO against removal by OH is about 50 days in the stratosphere (e.g. Mauzerall et al., 1998). Since the age of the plume after the injection date of 12 August 2017 (Peterson et al., 2018) was 24 and 25 days for the observation on 4 September and 5 September 2017, respectively, the mole fraction of CO in the plume was thus significantly affected by chemical loss due to the reaction of CO with OH. Therefore, the observed CO mole fractions are corrected by assuming a continuous removal by OH, an approach similar to the one used by Andreae et al. (2001), using the parameters presented by

Mauzerall et al. (1998).





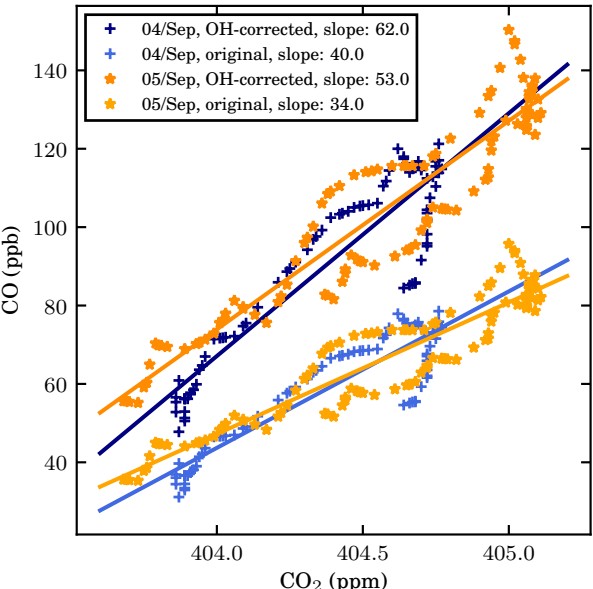

**Figure 4.** CO vs $CO_2$ scatter plot at the observed CO enhancement in the AirCore profiles from 4 and 5 September 2017. The CO and $CO_2$ data presented here lies between the pressure levels 167 hPa and 140 hPa (13.2–14.3 km) on 4 September and between 175 and 140 hPa (12.9–14.4 km) on 5 September. The OH-corrected CO versus $CO_2$ is also plotted.

The plume was transported dominantly in the stratosphere as shown by the back-trajectory analysis in Section 3.2. (Peterson et al., 2018) estimated the time of the up-draft in the troposphere to be around 5 hours. The dilution of the plume due to mixing with ambient air is ignored in the calculation. The OH-corrected enhancement ratios are plotted in Figure 4. The corrected enhancement ratios are $53 \pm 2$ and $62 \pm 3$ ppb ppm$^{-1}$ respectively. Thus, the loss of CO was about 35–37 %.

### 3.4 CO stable isotope composition

The LISA sampler CO mole fractions and stable isotope analysis results are presented in Table 4. Two types of samples can be distinguished, a plume sample and a background sample. It can been seen from Table 4 that the difference in their mole fraction and $\delta^{18}O$ is most pronounced.

The sample taken on 6 September can be considered as a background value for two reasons. First of all, the mole fraction measurements from both LISA and AirCore agree with those of normal NH CO stratospheric mole fractions (Hoor et al., 2005). Secondly, the $\delta^{13}CO$ agrees to within $\pm 1$ ‰ compared to the measurements performed in the southern hemisphere lowermost stratosphere (Brenninkmeijer et al., 1995). Note that tropospheric CO and its isotopic composition exhibit a latitudinal gradient and a seasonal cycle, related to the OH-sink. It is not exactly known whether and to what extent these gradients exist the stratosphere, an thus the agreement may be incidental.





**Table 4.** LISA observations of CO and $CO_2$ mole fractions and the carbon and oxygen isotopic composition of CO, of the plume, P, and background, B. Here $\delta^{13}C(CO)$ is reported in ‰ with VPDB as a reference material and $\delta^{18}O(CO)$ in ‰ with VSMOW as a reference material.

|  | Altitude (km) | $\theta$ (K) | CO (ppb) | $CO_2$ (ppm) | $\delta^{13}C(CO)$ ‰ | $\delta^{18}O(CO)$ ‰ |
|---|---|---|---|---|---|---|
| P (05 Sep) | 13.6 | 370.3 | 74 | 405.5 | -28.8 | 4.3 |
| B (06 Sep) | 13.4 | 368.9 | 34 | 405.2 | -29.6 | -1.0 |

The $\delta^{18}CO$ in the southern hemisphere SH is about 7.2 ‰ more depleted compared to the background value found from the observation presented here. Brenninkmeijer et al. (1995) attributed the relatively low values of $^{18}O$ to two determining factors; the unknown but probably low source signature for oxygen of methane-derived CO and the inverse kinetic isotope effect in the reaction with OH that depletes CO in $^{18}O$. Nonetheless, lower $\delta^{18}O$ values are typically a sign of absence of any nearby sources other than oxidation of atmospheric methane (Brenninkmeijer et al., 1999).

The LISA CO mole fraction measured on 5 September compares best to the AirCore measurement of the day before and is in the middle of the plume; see Figure 1. The isotopic composition of that particular sample is representative for that in the plume. A comparison of the different $\delta^{18}O$ values, see Table 4 clearly show that the CO plume is of different origin than stratospheric. Though the observed difference in $\delta^{13}C$ between the plume and the background sample is significant (0.8 ‰), it cannot be excluded that this is due to natural variability. Furthermore, many sources carry a comparable $\delta^{13}C$ signature, see

Table 2.

### 3.5    Source signature based on isotopic composition of CO

The observed CO enhancement carries a different isotopic composition compared to background air (Table 4). The additional information gained by isotopic analysis is useful since sources tend to produce products with a very distinct isotopic composition, which acts as a source signature.

Determining the source signature of the enhancement is not straightforward because of two reasons. First, the plume sample was affected by mixing in the troposphere during up-draft, and by mixing in the stratosphere. Secondly, the plume is too old to ignore oxidative loss of carbon monoxide. Thus, as explained in Section 2.3.2 the simple Keeling approach does not apply here.

        Both the tropospheric background in NH summer (Bergamaschi et al., 2001; Mak et al., 2003) and stratospheric background,
see Table 4, are more depleted in $^{18}O$ than the sample obtained in the plume. Thus, mixing would decrease the $\delta^{18}O$ of the plume. Since tropospheric and stratospheric $\delta^{13}C(CO)$ were alike (see Table 3), $\delta^{13}C$ is not obviously affected by mixing.

        In addition to mixing in the stratosphere, the time-scale of transport is long enough for significant removal by OH to be important. The isotopic composition of the plume is subjected to an inverse isotope effect at stratospheric pressure for both $^{13}C$ and $^{18}O$, depleting the remaining CO in both $^{13}C$ and $^{18}O$ (Röckmann et al., 1998). Thus, both removal by OH and





mixing make the plume CO more depleted in $^{18}$O; $\delta^{13}$C is mainly affected by OH. The plume isotopic composition was thus originally more enriched in both $^{18}$O and $^{13}$C.

Equation 4 is used to estimate the CO mole fraction and oxygen and carbon isotopic composition of the plume 25 days before the observations as a function of the unknown stratospheric fraction of air mixed into the sample, $f_{\text{strat}}$. It is assumed that pressure and temperature remained constant during transport. The number density of OH assumed for the stratosphere is

$1.2 \cdot 10^6$ and is as used to derive an estimate of the enhancement ratio in Section 3.3. The estimated CO mole fraction and the carbon and oxygen isotopic composition are shown in Figure 5 for different OH number densities. The model results show that the isotopic composition of the plume would have been $-27.5\,‰ \leq \delta^{13}\text{C} \leq -25\,‰$ and $10\,‰ \leq \delta^{18}\text{O} \leq 16\,‰$ after injection into the stratosphere, see Figure 5.

Using the OH-corrected estimate of $f_{\text{strat}} = 0.66$ from the Monte-Carlo simulation in Section 3.6, the plume $\delta^{13}$C and $\delta^{18}$O,

directly after injection in the stratosphere, are estimated to be $-25.6\,‰$ and $11.7\,‰$ respectively. Hence, the fractionation that occurred during transport depleted $^{13}$C by $3.2\,‰$ and $7.4\,‰$ $^{18}$O.

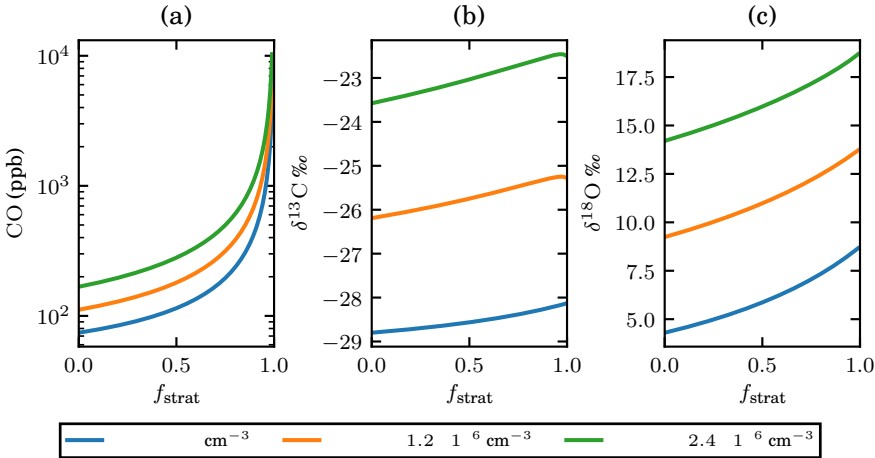

**Figure 5.** The estimated CO mole fractions (a), note the logarithmic scale on the y axis, and the carbon (b) and oxygen (c) isotopic composition of the plume CO for different OH number densities versus $f_{\text{strat}}$ just after injection into the stratosphere. OH number densities are shown in the legend, the value of $1.2 \cdot 10^6$ can be considered representative of the stratosphere. The value of $2.4 \cdot 10^6$ is arbitrarily added, as twice the stratospheric value and serves as an upper estimate.

Note that when the number density of OH is $0$ cm$^{-3}$ Equation 4 describes simple end-member mixing with rate $k_{\text{er}}$. If the Keeling method is applied, a source signature of $\delta^{13}\text{C}_{\text{src}} = -28.1$ and $\delta^{18}\text{O}_{\text{src}} = 8.7$ is obtained, suggesting a lower limit of the source signature. It can be seen from Figure 5 that even with very large amount of stratospheric mixing and high OH

number density the source signature has $\delta^{13}\text{C} < -22\,‰$ and $\delta^{18}\text{O} < 21\,‰$. Based on this analysis the source signature has $-27.5\,‰ \leq \delta^{13}\text{C} \leq -22\,‰$ and $10\,‰ \leq \delta^{18}\text{O} \leq 21\,‰$.





The analysis above shows that the plume was initially more enriched in both $^{13}$C and $^{18}$O than at the time of observation. Comparing this to the source signatures in Table 2, it is clear that the source signature is similar to that of CO produced in wildfires. Fossil fuel combustion sources, the only other source that produces CO containing higher $\delta^{18}$O, can be excluded

for two reasons. First, the source signature of high temperature combustion, $\sim 23.5$ ‰, would require an unusually high fractionation to explain the observed value. Secondly, the enhancement ratio of $\Delta$CO : $\Delta$CO$_2$ is too high for modern day fossil fuel combustion.

On the basis of the observed $\delta^{13}$C signature, CH$_4$-oxidation can be excluded as a source, as methane-derived $\delta^{13}$C is usually more depleted (Brenninkmeijer et al., 1999). Finally, oxidation of NMHCs can be excluded as a significant source. The total

amount of NMHCs in the stratosphere is on the order of several ppt (Scheeren et al., 2003). Furthermore, estimates of the NMHCs source signature suggests that the oxygen signature is in the range 0–3.6 ‰ (Brenninkmeijer and Röckmann, 1997; Vimont et al., 2019). The NMHCs produced in the fire have a mean enhancement ratio to CO of the order of ppt ppm$^{-1}$ (Mauzerall et al., 1998), and thus result in a very small in situ source of CO that has a very small effect on the isotopic composition.

### 3.6    Stratosphere-troposphere exchange estimate based on the in situ observations

The fractions of tropospheric, $f_{\mathrm{p}}$, and stratospheric air, $f_{\mathrm{s}}$, in the plume were determined using Equation 13 was used to determine the fractions of tropospheric and stratospheric air in the plume. The results of two Monte-Carlo simulations are shown in Figure 6, one simulation with CO observation corrected for OH, and one without the correction. The mode of each distribution suggest that the tropospheric air fraction is 46 % and the stratospheric fraction is 54 %. After the correction for

oxidation, this shifts the tropospheric contribution to 34 %, and the stratospheric contribution to 66 %. Thus, ignoring the oxidation results in bias in the estimated stratospheric and tropospheric contribution to the plume.

## 4    Discussion

In this study, it is shown that stable isotope analysis and mole fractions obtained by LISA and AirCore indicate the presence of wildfire smoke in the stratosphere. Using the stable isotope and mole fraction observation, the contribution of tropospheric air

and stratospheric air to the composition of the plume is estimated.

### 4.1    Enhancement ratios and plume age

Initially, the plume was observed from a clear CO mole fraction increase in the stratosphere, present in two AirCore profiles. The CO plume mole fraction measurements, up to 90 ppb, are lower compared to other plumes measured in the stratosphere, notably by Waibel et al. (1999) (300 ppb), Jost et al. (2004) (200 ppb), and by Cammas et al. (2009) (250 ppb). First, the plume

reported is older than other observations. The estimated plume age was 25 days, where the other observations were sampled after 7 to 14 days. Hence, the plume observed here was affected more by mixing and photo-chemistry in the stratosphere. Secondly, it is not possible to determine from the data presented here, whether the centre of the plume was sampled, in the

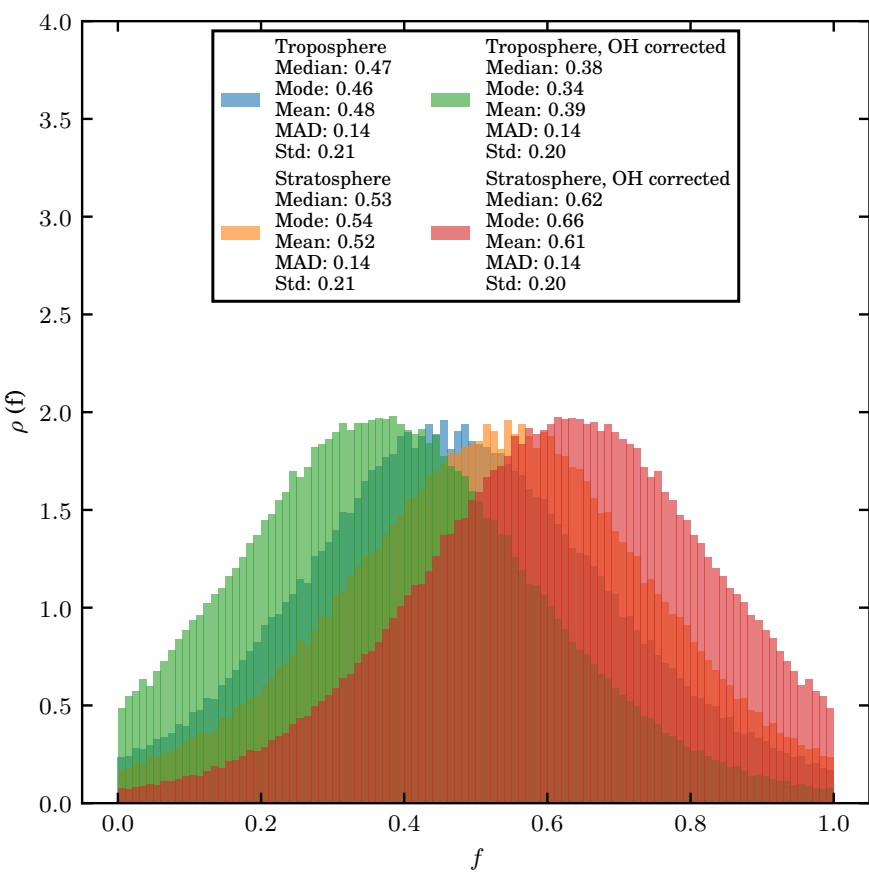

**Figure 6.** Probability density of the airmass fraction by volume, as a result of a Monte-Carlo simulation. Randomly selected input parameters, the end-member definitions presented in Table 3 were used to define Equation 13, which was consequently solved for the airmass fractions by volume $f$. This process was continued until $1 \cdot 10^6$ solutions were obtained, i.e. $1 \cdot 10^6$ solutions passed the filter (see Table 2.3.3). The data is binned into 1 % bins. A second run was performed with a correction for the oxidation applied to the observation, AP OH-corrected in Table 3. The mode of each distribution was obtained from a fit to the distribution.





horizontal sense, where CO mole fractions are highest. Finally, it should be pointed out that the plume was encountered during AirCore vertical profiling which has limited vertical resolution. The vertical resolution of the AirCore measurements was

estimated to be about 150 m between 13 and 14 km.

In addition to the anomalous CO mole fraction, enhancement ratios of CO vs $CO_2$, based on concurrent AirCore $CO_2$ and CO measurements, agree with the ratios measured previously in biomass burning emissions. The observed ratios of 34–40 ppb ppm$^{-1}$ in this study is lower than the measured enhancement ratios of 50 ppb ppm$^{-1}$ by Jost et al. (2004) and 48–73 ppb ppm$^{-1}$ by Andreae et al. (2001). The plumes reported by both Jost et al. (2004) and this study originated from

forest fires in North America (40°–55° N), and were observed at a similar altitude of approximately 1.5 km above local tropopause. It is a reasonable assumption that the initial enhancement ratios and the mixing of the plumes with lower stratospheric backgrounds are similar. However, the plume age of roughly 25 days in this study is significantly older than that of 10–14 days observed by Jost et al. (2004). Therefore, the ageing of the plume coupled with the OH-related destruction of CO likely explains the difference in the observed ratios. Similarly, the age of 9–10 days of the plumes observed by Andreae et al.

(2001) is also significantly younger than the plume age found in this study. Furthermore, their observation was made in tropical tropospheric air, and the background CO mole fraction of about 100 ppb is significantly higher than the stratospheric background of roughly 40 ppb, which contributes to their higher observed enhancement ratios than those in this study. Indeed, the OH-corrected enhancement ratios of 53–62 ppb ppm$^{-1}$ in this study come closer to the similar OH-corrected enhancement ratios of 64–98 ppb ppm$^{-1}$ in Andreae et al. (2001), and the remaining difference may be caused by the different type of fuels

of biomass burning, forest in this study and savannah/forest in Andreae et al. (2001).

This ratio is comparable to the measured high-altitude enhancement ratio, 50 and 55 ppb ppm$^{-1}$ Jost et al. (2004) however lower than those observed by Andreae et al. (2001). They found 74 ppb ppm$^{-1}$ for a higher altitude plume after correction for plume ageing. There are two important differences between the measurements in this study and the measurements presented by Andreae et al. (2001). First, their observation was made in tropical tropospheric air, which is affected by different photo-

chemistry and transport properties than the stratospheric measurements presented here. Secondly, they found a lower age of 9.5 days for the observed plume which makes the correction for removal by OH less uncertain.

## 4.2   Isotopic composition of plume-CO

The stable isotope source signature of CO qualitatively supports wildfire smoke as the source. Very little stable isotope measurements exist on stratospheric air. On the other hand, several tropospheric measurements of other wildfire events were pub-

lished so a comparison can be made with those.

First, the $\delta^{18}$O source signature for Siberian boreal forest fires, 14.8 ‰, and 9.0 ‰, respectively (Bergamaschi et al., 1998; Tarasova et al., 2007), compare well with the observation made in this work after estimating the fractionation that occurs during oxidation, see Section 3.5. Although the transport-history of the observations presented here complicates the determination of the source signature, it was shown that the original $\delta^{18}$O was higher.

In addition, Brenninkmeijer and Röckmann (1997) found a source signature of 4.5 ‰ for wildfire smoke, based on southern hemisphere observations. As argued by Brenninkmeijer and Röckmann (1997) the samples must have been affected by the



strong fractionation accompanying the reaction of CO with OH along the mixing, something that was not accounted for in their method used to derive the source signature. The lifetime of CO against removal by OH in the stratosphere is considerably longer than in the troposphere, and hence the wildfire sample presented here was likely less affected by fractionation than their
measurement.

It is shown that CO stable isotope measurements can help pollution events in the stratosphere to be identified. It must be noted that this study took advantage of the fact that, on the days following the pollution event, clean background air was sampled. Thus, a direct comparison of background air and pollution air was possible. Without the measurement of background air, source attribution would have been difficult from stable isotope measurements, as little is known about the CO isotopic
composition. Fundamental knowledge of CO isotopic composition and its temporal and latitudinal variation in the stratosphere is vital for the detection of future pollution events based on CO measurements.

In addition to a poorly understood isotope budget of the stratosphere, studies would benefit from measurements of source signatures from lab experiments. Although methane-derived CO can be discriminated based on $^{13}$C, most of the other sources are similar in $^{13}$C. The case made in this study would have been stronger, if the oxygen signature of both methane and NMHCs
were known more precisely. Our fundamental knowledge of the CO isotopic composition in the stratosphere would also benefit from those measurements, as methane is the main source of CO in the stratosphere.

### 4.3 Assessment of tropospheric and stratospheric airmass contributions

Finally, the airmass-fraction of the troposphere and stratosphere were derived using the tracer observations. The results suggest that the 2017 pyro-Cb plume observed above Sodankylä consist of approximately $34 \pm 14$ % tropospheric air polluted with
wildfire smoke. This is in qualitative agreement with model simulations from Trentmann et al. (2006) on the Chisholm fire in 2001, an event similar to the 2017 British Colombia fires. Yet another event was modelled by Cammas et al. (2009) and they estimated the amount of polluted boundary layer air above the tropopause to be 15–20 %.

### 5 Conclusions

A wildfire smoke plume in the lower stratosphere is investigated using in situ observations CO and $CO_2$ from AirCore, and
stratospheric $\delta^{13}$C and $\delta^{18}$O in CO from LISA. The plume was identified by enhanced CO mole fractions at approximately 13.6 km altitude, present on two consecutive days. The plume's enhancement ratio of CO to $CO_2$ mole fractions was in the range 34–40 ppb ppm$^{-1}$. The stable isotopic composition of carbon and oxygen in CO support wildfire smoke as the source for the enhanced CO mole fractions observed in both AirCore and LISA samples. Using the CLaMS back-trajectory module and CALIOP backscatter data the source region is determined to be British Colombia, Canada. The smoke was injected on
12 August 2017, 24–25 days before the observations were made. The age of the plume aided in the estimation of the amount of oxidation, a 35–37 % loss of CO, and the accompanying isotopic fractionation, 3.3 ‰ for $\delta^{13}$C and 7.6 ‰ $\delta^{18}$O. Using this information, the enhancement ratios corrected for oxidation ranged from 53 to 62 ppb ppb$^{-1}$. The plume isotopic composition of oxygen and carbon in CO was estimated to be $-25.6$ ‰ and 11.7 ‰. From the LISA observations, it was possible to





determine the fractions of tropospheric, $34 \pm 14$ %, and stratospheric air, $66 \pm 14$ %, in the plume using a three end-member

mixing model.

**Appendix A: Derivation of Equation 6 and Equation 8**

A constant entrainment rate is assumed, i.e. so that

$$\frac{dV}{dt} = kV \tag{A1}$$

This can be solved to yield:

$$V(t) = V_0 \exp(kt) \tag{A2}$$

with $V_0$ the initial volume of the air-parcel containing the contamination. Thus $V_0/V_t = f_{\text{plume}}$ and

$$V(t) = f_{\text{plume}} V_{\text{plume}} + f_{\text{strat}} V_{\text{strat}} \tag{A3}$$

with $f_{\text{plume}} + f_{\text{strat}} = 1$. Then Equation A2 can be written as follows:

$$1 - f_{\text{strat}} = \exp(-kt) \tag{A4}$$

this can be rearranged to give $k$ in terms of $f_{\text{strat}}$:

$$k = \frac{-\ln(1 - f_{\text{strat}})}{t} \tag{A5}$$

Starting from

$$\frac{dn(\text{CO})}{dt} = -k_{\text{er}}(n(\text{CO}) - n_{\text{bg}}(\text{CO})) - k_1 n(\text{OH})n(\text{CO}) \tag{A6}$$

Letting $n(\text{CO}) = x$ and:

$$a = -k_{\text{er}} - k_1 n(\text{OH}) \tag{A7}$$

and

$$b = k_{\text{er}} n_{\text{bg}}(\text{CO}) \tag{A8}$$

After substitution and rearranging would result in:

$$\frac{dx}{ax + b} = dt \tag{A9}$$





Integration yields

$$\frac{1}{a}\ln\left(ax+b\right)+C=t \tag{A10}$$

where C is an integration constant. Solving for $x$ gives:

$$x(t)=\frac{1}{a}\left(\exp\left(at\right)\exp\left(-Ca\right)-b\right) \tag{A11}$$

$\exp-Ca=$ constant so it can be replaced by yet another arbitrary constant $c$.

$$x(t)=\frac{1}{a}\left(c\exp\left(at\right)-b\right) \tag{A12}$$

After doing so, the constant $c$ can be determined from the boundary condition $x(t=0)=x_0$:

$$x_0=\frac{c-b}{a} \tag{A13}$$

which is equivalent to:

$$c=ax_0+b \tag{A14}$$

substituting Equation A14 into Equation A12 yields:

$$x(t)=\left(x_0+\frac{b}{a}\right)\exp\left(at\right)-\frac{b}{a} \tag{A15}$$

Finally Equation A7 and Equation A8 and $n(\mathrm{CO})=x$ can be used to obtain:

$$n(\mathrm{CO})(t)=\left(n_0(\mathrm{CO})+\frac{k_{\mathrm{er}}n_{\mathrm{bg}}(\mathrm{CO})}{-k_{\mathrm{er}}-k_1 n(\mathrm{OH})}\right)\exp\left(-\left(k_{\mathrm{er}}+k_1 n(\mathrm{OH})\right)t\right)-\frac{k_{\mathrm{er}}n_{\mathrm{bg}}(\mathrm{CO})}{-k_{\mathrm{er}}-k_1 n(\mathrm{OH})} \tag{A16}$$

*Author contributions.* JJDH and RK performed the fieldwork. JJDH and MEP performed the stable isotope measurements. JUG, IT, and
RM did the back-trajectory analysis. HC retrieved the AirCore profiles. JJDH and HC wrote the manuscript with contributions from all
co-authors.

*Competing interests.* There are no competing interests present.

*Acknowledgements.* The authors thankfully acknowledge the staff from FMI Arctic Space Centre for their efforts in balloon launching
and payload retrieval. This work is supported by NWO grant ALW-GO/15-10. The AirCore balloon flights are supported by ESA project
FMR4GHG. We tahnk the ECMWF for providing the reanalysis data.



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
