# Peer review of "Wildfire smoke in the lower stratosphere identified by in situ CO observations"

_Atmospheric Chemistry and Physics, 2020_

## Referee Comment (RC1) · Anonymous Referee #2 · 13 Apr 2020

**GENERAL COMMENTS**

The paper "Wildfire smoke in the lower stratosphere identified by in situ CO observations" presents measurements of CO mole fractions, CO2 mole fractions, and isotopic composition of CO from two balloon-borne instruments. Overall, the paper presents new datasets and includes in depth analysis of the measurements. It is generally wellwritten, with clear discussion of results. The paper would make a good contribution to ACP, after the following comments are addressed.

For the most-part, I found the description and justification for the methods chosen were complete and well-referenced. However, I do not have the expertise to fully evaluate some details of the data collection and analysis methods (Sect. 2.1, 2.2 2.3.2, 2.3.3). In some sections, I found it a bit difficult to follow the methods. For example, some

descriptions of the methodology are written into figure and table captions. Throughout the paper, it would be helpful if high level information was provided at the beginning of each section to help guide the reader and explain how and why certain methods were applied. I have included some specific comments below, which include examples where additional descriptions would help.

**SPECIFIC COMMENTS:**

Line 15: "The in situ observations provide information... of the 2017 smoke plume" The closing sentence of the abstract is a bit unclear. Is the "information on the trace gas chemistry" what was described in the previous part of the abstract? If so, perhaps could use more specific language (e.g., what new information is provided by this study?). Also, I don't see much mention of the 1 km width of the plume in the text (I assume this inferred from Sect. 3.1?). If this is a key result, then maybe it warrants more discussion in text.

Line 59 "Methods": A very short overview of what the AirCore vs LISA measure (mole fractions vs isotopic composition) and to what approximate vertical / temporal resolution would be helpful here.

Line 139: I found it very difficult to understand the method used for the back trajectories until I reached Fig 2 and the associated text in Sect. 3.2, which walks the reader through this process. Until reaching this figure, I found terms like "piece-wise manner" confusing. To remedy this, the authors could merge Sect. 2.3.1 directly into Sect. 3.2 or perhaps make Sect. 2.3.1 a bit more general with a forward reference to Sect. 3.2 (e.g., something like used back trajectories from CLaMS and a piecewise method illustrated in Sect. 3.2).

Line 148: A bit of an overview would be helpful for Sect. 2.3.2 and 2.3.3. How are the methods being applied to the data? What information do the methods in Sect. 2.3.2 provide compared to Sect. 2.3.3? Have these methods been applied to similar datasets before?

**ACPD**
Line 152: Did not define "f" in the text. Check throughout that all variables in equations are defined.

Line 191: I found this section confusing at first. I had to jump around the text to understand how Table 3 was constructed and how this information was used. Some of the methods/categories are only really described in the Table 3 and Figure 6 captions. Also, the Table 3 caption references "Monte-Carlo simulation" at the very top of the caption, but the MC simulation is not mentioned until much further in the text. One possible way to improve this would be to give a high level overview when the table is first introduced. This would describe why the table was put together, how the various lines of the table were compiled (with cross-references to relevant sections), the purpose / input parameters of the MC simulation when the table is first introduced.

Table 3 caption: The caption is a bit confusing –review/rewrite and maybe move some of the details into the text describing the methods. For example, "... does not significantly affect the results" – the results of what? The MC simulation?

Table 3: Where did the numbers for AP OH-Corrected come from? Are these related to the numbers given around line 337 – if so, how exactly were they chosen (they don't seem to exactly match any of the numbers given in the text)?

Line 237: "This thus differs... in Table 2". Could you clarify this statement? It looks like the values in Table 3 fall within the range of Table 2 – but are a narrower part of the range (because from specific type of biomass burning?)

Line 240: Please add a bit of a description of the Monte-Carlo simulation. It's not clear what data is being used, what equation it's applied to, and what the desired output is until the reader gets to the caption of Fig 6.

Line 255: "The observed CO2... Which allowed for determination of enhancement ratio" Is the observed CO2 difference significant for both LISA and AC? It looks like LISA is biased high compared with AC (e.g., non-enhanced day for LISA is comparable
to biggest enhancement for AC). Also, why does the slight increase in CO2 allow for enhancement ratio calculations?

Line 259: This section could use an opening sentence describing what the back trajectories are being used for.

Line 260: What is the "match distance"? Is this the minimum distance from the back trajectory to the CALIOP scan? Also, since the centre/upper altitude are not shown – is this the lower altitude? Is this starting from 13.3 km (based on Sect. 2.3)?

Figure 2 caption: "...the correspondence was sufficiently good". Does this mean the altitude correspondence between the back trajectory and CALIOP aerosol enhancement?

Line 277: How are the slope / uncertainty calculated?

Line 307: How do you know that the plume is clearly not stratospheric? Or is it just clear that the plume is different from background?

Line 332: How did you choose the ranges of possible values from Fig 5?

Figure 5: The numbers in the legend aren't showing up correctly. (No value for the blue marker, reading at 16 instead of 106 for the orange and green markers)

Line 343: Why is it within the range for "biomass burning" in Table 2 but not within the range for "wildfire smoke" in Table 3?

Line 414: You mention that "little is known of the CO isotopic composition". Do your measurements of the background air mass contribute to this knowledge? If so may be worth mentioning the background measurements in the abstract.

Line 426: "Yet another event was modelled..." is this another wildfire event? If so, specify.

**TECHNICAL CORRECTIONS:**
Line 10: "Back-trajectory analysis, performed with.... Date of 12 August 2017" Very long sentence. Consider breaking it up.

Line 13: "Colombia" -> "Columbia" (also, line 426, 434)

Line 145: "Wherever the distance was below", replace with "Wherever the distance was smaller than"

Line 245: "its" -> "it is"

Figure 2 caption: Check capitalization after periods. Also, last sentence is run on "CALIOP, the time" -> "CALIOP. The time"

Line 307: ", see Table 4" -> "in Table 4"

Line 308: "stratospheric" -> "the stratosphere"

Line 309: "it cannot be excluded" -> "it cannot be ruled out"

Line 356: delete repetion? "... was used to determine fractions of tropospheric and stratospheric air in the plume"

Line 414: "pollution air" -> "polluted air"

---

## Author Comment (AC1) · 16 Sep 2020

**Reply to interactive comments on "Wildfire smoke in the lower stratosphere identified by in situ CO observations", by Joram J. D. Hooghiem et al.**

September 16, 2020

**General**

We thank the two reviewers for their feedback on our manuscript. The manuscript has been improved by making use of their suggestions, most notably those concerning the methods section. Our replies can be found in blue, below specific comments of the two separate reviewers. In places, the revisions from the manuscript are copied, and depicted in boldface.

We should mention that we have found a mistake in one of our calculations that led to errors in several places in the manuscript. It concerns the mole fraction and isotopic composition calculations discussed in Sections 3.5 and 3.6. It does not affect the general conclusions. The ranges in 3.5 changed from $-27.5$ ‰ $\leq \delta^{13}C \leq -25.0$ ‰ to $-27.5$ ‰ $\leq \delta^{13}C \leq -26.6$ ‰ , and 10 ‰ $\leq \delta^{18}O \leq 16$ ‰ to 10 ‰ $\leq \delta^{18}O \leq 14$ ‰ . The estimated fraction of tropospheric air in 3.6 changed from 34 to 45 %.

In addition to the changes discussed below, a few additions to the manuscript have been made. A panel has been added to Figure 3 that shows the complete trajectories; this panel aids in the explanation of the methods. Also, the tropopause altitude is now added to Figure 1.

**General Comments Referee #1**

The paper "Wildfire smoke in the lower stratosphere identified by in situ CO observations" presents new data collected in the stratosphere by two separate sampling systems. The scientific merit is high, and the analysis methods are sound. In particular, the authors have presented an algorithm for assessing the impact of OH on CO during transport to the stratosphere. The paper overall is well structured. However, at times the sentence structure and descriptions of the methods are confusing, and several key areas in the results and discussion sections lack needed clarity/explanations. Overall, the paper is excellent, and will make a great contribution to ACP after several minor revisions (see below). Overview of Revisions Needed:

1. In section 2.2.2 and 2.2.3, the LISA sample transfer and CO isotopic measurements are described. Hooghiem et al. (2018) and Pathirana et al. (2015) are cited for reference on the LISA sampler and the CO isotopic analysis line, respectively. However, I was unable to find tests in either reference that show storage tests performed on the sampling containers used in this study. The authors make special note to point out the storage flasks are sealed with Viton O-rings, which is known to contaminate for CO mole fraction on the order of 1-2 ppb per day (e.g. Novelli, Steele, and Tans, 1992). A reference or supplemental section should be included that describes laboratory tests which demonstrate that $x(CO)$, $\delta^{13}CO$ , and $\delta^{18}CO$, are unaffected by a 7-month storage time in flasks with a Viton oring. This is especially important given the low mole fractions you have observed in the stratosphere as well as the few data points

you present here. Any sampling bias could have a large impact on your source attribution.

A stability test was performed using stratospheric air samples obtained from the LISA flights over Sodankylä in 2018. Analysis, transfer, and storage were performed using the same methods as presented as in the present study. Re-analysis on 4 samples was performed using the same Picarro instrument in July 2020. The drift in $x(\text{CO})$ was computed to be 0.05 ppb day$^{-1}$, assuming a linear drift.

The stability of the $\delta^{13}\text{C(CO)}$ and $\delta^{18}\text{O(CO)}$ in these flasks was not assessed directly. The flasks used for storage have a two O-ring configuration which is known to give better results in storage test (Sturm et al., 2004), this is added to the methods section: **"The flasks have a Rotulex connection and are sealed with two Viton-70 O-rings, providing better sample stability than single O-ring configuration (Sturm et al., 2004)."** As discussed in text, the stratospheric background measurement compared well to other stratoshperic measurements, and the plume sample differs significantly.

A computation is provided to show the likely magnitude of a contamination. The source of contamination shall be the air surrounding the flasks. Assuming that the contamination can be modelled using the mixing model provided in the manuscript, e.g. Eq. 1 and 2. Atmospheric measurements on northern hemisphere background air from Mak et al., 2003 provides values $x(\text{CO}) = 100$ ppb, $\delta^{18}\text{O} = 0$ ‰, and $\delta^{13}\text{C} = -29$. The estimate results in no observable difference in $\delta^{13}\text{C}$, and $-0.7$ ‰difference for the plume in $\delta^{18}\text{O}$. Alternatively, more polluted, air values like one would find in spring $x(\text{CO}) = 150$ ppb, $\delta^{18}\text{O} = 4$ ‰, and $\delta^{13}\text{C} = -26$. Also leads to less than 0.2 ‰for both $\delta^{13}\text{C}$ and $\delta^{18}\text{O}$ values of both the plume and background.

In this computation, fractionation is ignored entirely. Also, the computation is very sensitive to the contamination source values and drift rate. Here we tried to give a plausible estimate based on atmospheric values. It should be noted that a drift in mole fraction does not necessarily mean a change in isotopic composition (as illustrated above) nor does a stable mole fraction guarantee a stable isotopic composition during storage. A systematic bias cannot be excluded, but may be small. We have explicitly stated in the method section that we cannot exclude a bias due to the long term storage:

**"Yet, it can not be excluded that the isotopic measurements, see below, are biased by more than a ‰."**

2. The analysis of the enhancement $\text{CO}:\text{CO}_2$ ratios presented in the results and discussion is confusing as it is presented currently. In the methods section, the authors describe a common method for obtaining enhancement ratios: assuming that an air parcel is a mixture of some sources plus a background. The authors further show that they have made good background measurements for the altitudes of interest in the stratosphere on Sept 6 and 7, at least for mole fraction. However, in the analysis, particularly in Figure 4, it does not seem that any background values have been subtracted off. The enhancements

presented in figure 4 appear to just be the mole fraction measurements from 4 and 5 September in figure 1. In figure 4, the authors appear to be attempting to find a slope to predict the overall CO:CO$_2$ ratio in the plume. The x axis should range from 0 to 1 (or possibly 1.5) ppm, consistent with the 6th and 7th September xCO$_2$ curves (background) being subtracted from the 4th and 5th September xCO$_2$ curves. The xCO enhancement on the y-axis should similarly be adjusted. Given that the background curves are not constant through the altitude range considered, I think that the subtraction may alter the results of the slopes presented in figure 4. This is, of course, provided that the error is not simply a typographical error in the labeling of the axis. If, on the other hand, the authors are attempting only to show the CO:CO$_2$ ratios of the plume, and not subtract the background, then the methods should be clarified.

A description of the methodology applied was missing. Two paragraphs have been added to the methods section to clarify the methodology:

**"CO and CO$_2$ are co-produced in burning processes, and their emissions into the atmosphere result in an enhancements, $\Delta$CO and $\Delta$CO$_2$, compared to background air. The enhancement ratio of $\Delta$CO : $\Delta$CO$_2$ is typically high for wildfires, and decreases over time due to photochemical loss of CO (Mauzerall et al., 1998). The enhancement ratio is conserved during mixing, if the background is constant. Thus, in this case, the enhancement ratio can be directly obtained as the slope of a linear regression performed on AirCore CO and CO$_2$ data for the two separate AirCore flights that sampled the plume on 4 and 5 September 2017."**

**"Alternatively, measurements of background air can be used to compute the enhancement ratio directly. It is assumed that the flight performed on 6 September is representative for the background air adjacent to the plume. CO and CO$_2$ data are smoothed using a moving average with an averaging window of 25 data points in order to reduce the analyser noise in CO. The background air is interpolated on isentropic surfaces to the observed plume altitude, and the enhancements are computed directly on isentropic surfaces. As we will see, the total enhancement in CO$_2$ is small, and only the results with $x$(CO$_2$) > 0.2 ppm are used in the subsequent computation of the enhancement ratios."**

The enhancement ratio is preserved if a background has constant mole fractions in the tracers for which the enhancement ratio is calculated. So the implicit assumption made in the manuscript, was that the background is constant. If the background is constant, it does not matter which dimension is used to intersect the plume (i.e. horizontal or vertical). Moreover, the actual background mole fractions need not to be known in order to determine the enhancement ratio. Subtracting a constant value for both $x$(CO$_2$) and $x$(CO) respectively would merely result in a change of the intercept of the linear fit applied. The intercept is not of interest. Hence, when the assumption of a constant background can be justified, the enhancement ratio of the plume is the same as the slope of a tracer-tracer correlation. The justification is provided by the overall regression

statistics, the $r^2 > 0.83$ and the 95 % confidence interval on the slope.

We agree that validity of the assumption of constant background is questionable. This is now discussed together with an additional computation of the enhancement ratios (see added paragraph above). The discussion is updated with the following paragraph

**"The enhancement ratios computed directly are much higher than the results discussed above. First, it is difficult to assume what background should be used. The stratospheric background mole fractions, especially the $CO_2$, varies with altitude and in time, e.g. comparing the AirCore profiles from 6 and 7 September, which questions the assumption of a constant background. It is thus difficult to state with certainty that data from 6 September is representative for the background. However, as the plume, and air directly adjacent to it, move together, it can be assumed that the air surrounding the plume has constant mole fractions. The best estimate of the mole fractions are those measured above and below the plume, obtained from the vertical profile. Hence, the most reliable estimate of the enhancement ratio is obtained from the regression analysis discussed above. On a final note, the very small standard deviation of $1$–$2$ ppb obtained for 4 September is largely due to the small amount of data, $N = 5$, and the large averaging window of 25 data-points."**

3. Lastly, the LISA flight from the 4th of September is mentioned in the methods section, but I have been unable to find this data in the results and discussion of the isotopic measurements. Why is this missing? It would further strengthen your source attribution section as you currently only have one plume and one background point. If it was discarded, the reasoning should be presented. I recommend that you add this point into your dataset if possible.

It is correct that no data is presented here from the LISA flight of September 4. In the methods a more generic description of data retrieval is presented. Data collected from LISA on 4 September has no further relevance to this study, since it didn't collect samples in the lowermost stratosphere. The date of 4 September is mainly relevant for the AirCore measurements of which contains the plume measurement. A line has been added into the methods that clarifies which data from LISA is used in the presented study.

**"The AirCore was analysed for $CO_2$, $CH_4$, and CO mole fractions, for details see Section 2.2.1. In this work only the AirCore $CO_2$ and CO profiles are used. LISA samples have been analysed for $CO_2$, $CH_4$, and CO mole fractions, see Section 2.2.1, and the stable isotopic composition of CO, see Section 2.2.3. Here only the LISA CO mole fraction and CO isotope measurements at the plume altitude, see Section 3.1, are used, one sample from 5 and 1 sample from 6 September 2017. Although measured, LISA $CO_2$ appears to suffer from a bias as concluded from comparison with AirCore measurements, see Hooghiem et al., 2018."**

**Specific Comments Referee #1**

Pg 1, Ln 16: 1km? Into the stratosphere or you only sampled 1 km of the plume? Clarify.

The plumes vertical extent is 1 km, this has been clarified in the revised version.

**"Finally, the plume was extending over 1 km in altitude, as inferred from the observations."**

Pg 2 Ln 24: Specify dates.

The dates of the lidar observations have been added.

**"In 2017, a large smoke plume in the stratosphere was observed on several days between 24 August and 26 September by ground-based LIDAR and the Cloud-Aerosol Lidar with Orthogonal Polarization (CALIOP) aboard the CALIPSO satellite (Khaykin et al., 2018)."**

Pg 2 ln 47: You say that Mauzerall et al. 1998 also measured $CO:CO_2$ just before stating that only Jost et al measured $CO_2$ allowing for $CO:CO_2$. Please reword/clarify this statement.

It was meant to say, that Jost et al presented the only observation of the enhancement ratio in the stratosphere. This is clarified.

**"Of the stratospheric observations, only Jost et al., 2004 measured $CO_2$, allowing $\Delta CO : \Delta CO_2$ to be quantified, confirming the smoke's origin."**

Pg 3 ln 69-71: These three sentences need to be reworded. They are awkward to read. Explain what you mean by the sample is limited. I.E. explain how the low pressure and high rate of descent in the stratosphere results in 0.3L air for xxx m of stratosphere, compared to 1.1L of air for xxx m of troposphere. The low resolution and small sample volume lead to the need for LISA, etc...

The sentence have been revised for a better explanation.

**"The AirCore has a volume of 1.4 l. The AirCore takes a sample passively, and due to the low pressure in the stratosphere only about 0.3 l of the sample is stratospheric. The AirCore has a vertical resolution of 374 m at 200 hPa or 12 km altitude."**

Pg 3 ln 73: Larger amount relative to AirCore? You specify the L STP for AirCore, you should state the same here for LISA.

We have added the total amount of stratospheric air that LISA samples for a direct comparison.

**"The active sampling results in a larger amount of sample 180–800 ml per sample, thus allowing for isotope analysis."**

Pg 3 ln 75: Compare to AirCore here as well.

We have added the vertical resolution of 374 m at 12 km altitude of the AirCore used in this study (see reply to the comment about "Pg 3 ln 69-71" above.

Pg 3 ln 80: "In addition to . . . " Combine these two sentences

The last 2 sentences have been combined. The contribution of the radiosonde is now summarized in one sentence.

Pg 3 ln 87: Explain why two different instruments and models for the two analyses?

Two instruments were used in order to reduce time between payload retrieval and analyses for both AirCore and LISA in the field. The model only matters in the sense that they both measure $CO_2$, $CH_4$, CO, and $H_2O$. We have added this to the revised manuscript.

**"Two analysers were used, to allow for simultaneous analyses of both AirCore and LISA samples after payload retrieval (analyser models used: Picarro G2401 (AirCore), Picarro G2401-m (LISA) see e.g. (Crosson, 2008) for more information on the CRDS-analyser)."**

Pg 3 ln 84 and throughout text: Note: mole fractions to be represented by lower case x with subscript chemical formula per IUPAC standards. I personally do not have a problem with the nomenclature used in the text, but the authors may wish to make the change for technical correctness

The manuscript, both revised and original, were intended to be fully compatible with definitions by IUPAC (Cohen et al., 2007) at least as far as equations go. For example Equation 1, correctly uses italic $x$ for the mole fraction with upright text to specify its origin. In text we believe "$CO_2$ mole fractions" as well as its symbolic representation "$x(CO_2)$" are allowed, just as you would say pressure measurements in air vs $p$ measurements in air. Note that footnote two, chapter 2.10 on page 47 of Cohen et al., 2007, states: "... When the chemical composition is written out, parentheses should be used, $n(O_2)$", which also applies to mole fractions, according to page 48 of Cohen et al., 2007.

Pg 4: Section 2.2.3: I am not seeing the number of samples you have measured. You state above that you flew LISA on 3 flights, with a total of 4 samples per flight. However, you only present 1 data point per day. Please explicitly state the number of flasks you analysed for stable isotopes and if you averaged them for a given day. This is especially important below when you do not show the 4th Sept data (See further comments below and above on 4-Sept data).

This has been clarified. See also our reply to main comment number 3.

Pg 5 ln 124: Replace "concentrations" with "mole fractions".

We have replaced "concentrations" with "mole fractions".

Pg 5 Ln 135-138: It is unclear here what aspects of the CLaMS you used? This sentence seems to indicate that you used the advection scheme but not the mixing scheme, while at the same time implying that both the advection and mixing schemes are needed to resolve 2d filamentary structures. Please clarify.

We only used the trajectory module of CLaMS. A complete reconstruction of the 2-d (or even 3-d) structures of the exhaust would be much more complicated. Also, little would be known about the vertical velocity. Therefore we added the comparison with the CALIOP data from which we could piecewise trace the observed cloud. We have separated generic information from what this work uses

**"In the present study, only the advection scheme is used. First, information on the aforementioned additional vertical motion is lacking. Secondly, only synoptic scale transport is of interest to determine the source region of the smoke."**

Pg 5, ln 147: How accurately? This paragraph does not indicate the scale to which the smoke plume can be traced using this method. Suggest removing this word.

This word has been removed.

Pg 6 ln 153: "similar balance" should be "similar mass balance".

Changes implemented as suggested by the referee.

Pg 6 ln 155: Suggest modifying equation (2) to include "approx. equal" since delta notation here is a good approximation for small changes in delta, but strictly speaking, there is loss of tracer (very small) in this approximation (see Tans, 1980).

The $\approx$ is adopted in the revised manuscript, and Tans, 1980 is added as a reference.

Pg 6, ln 170: Remove the sentence "The production . . . neglected". This is confusing here and is not relevant to this section which is explaining the oxidation of CO by OH. This can be explained further in the Discussion section about how you determined the actual source.

The sentence is removed, as suggested.

Pg 8 Ln 200: "Volumetric Air Fraction": this is a confusing term. Looking at your equations, f is the fraction of the total mole fraction. You have a measured mole fraction for your given air parcel that you are trying to partition with your model. Furthermore, you are looking at CO here, not air, and volumetric is

additionally confusing as you have introduced no volumes in this equation that I can determine. Please clarify or restate.

*f* is the fraction of molecules coming from different sources substantiated by the subscripts (tropospheric, stratospheric, or wildfire). We have added a clarification for the parameter f defined in text. From the ideal gas law it follows that at constant temperature and pressure f can be computed equivalently from volume or from number density. Yet, we agree that the use of the term "volumetric" might be more confusing. Therefore we have redefined it to fraction of molecules.

**"Here $f_{bg}$ is the fraction of molecules of bg in ap, and similar for src"**

Pg 8- Pg 9, Ln 216-219: This needs more explanation. Please expand or clarify. This currently leaves the reader with the impression that your signal to noise ratio is something like 1:1, which would mean that you could not differentiate between the measurement noise of your measurement system and real atmospheric signals. This is certainly not the case for the AirCore. Also, please explain/expand on why 2x the measurement uncertainty was chosen and why this is assumed to be valid. The current level of explanation makes this choice seem arbitrary.

Pg 10 Ln 245-247: As with my previous comment on this section, you need to clarify and explain more clearly how you are assigning uncertainty, how you are interpreting your results within the bounds of that uncertainty. Why are you limiting your results to live within the 1 sigma bounds, for example? I'm not suggesting this approach is wrong, but it certainly needs further explanation.

This reply concerns the two previous comments together. The methodology has been updated with a better clarification and argumentation, as where the inputs of the Monte-Carlo simulation come from, including the chosen uncertainties.

**"The stratospheric end-member definition and the plume observation are based on the balloon-borne observations by LISA, outside and inside the plume, respectively, see Table 4. The plume mole fractions are assigned an uncertainty equal to the measurement uncertainty of LISA. The uncertainty of the stratospheric end-member is also a good measure of the variability surrounding the plume based on AirCore measurements between 12 and 14.5 km of 6 and 7 September. The uncertainty of the isotopic composition in the Monte-Carlo simulation is set equal to twice the measurement uncertainty (see Section 2.2.3) because the stratospheric variability of the stable isotopic composition of CO is unknown. Secondly, there may have been a small drift in the isotopic composition."**

Figure 1: Why is 4-Sept LISA measurement missing from this plot?

It is not missing, on that day there was no sample in the shown altitude

range. Note that with the updated methodology, it is made clear that LISA data from 4 September is not used throughout this work.

Pg 13 Ln 276-279/throughout section 3.3/Figure 4: Figure 4 and the description presented here are confusing (see General comment 2 above). You refer to enhancements and enhancement ratios, and in Section 2 you describe the source attribution in terms of sources on top of a background, or wildfire plume injection on top of the background stratosphere. However, in this section, it is not clear where you have subtracted your background or what you are using for a background. Are you using the profiles from the 6th and 7th averaged to assume a "typical" stratospheric background CO and $CO_2$? Furthermore, figure 4 would be clearer if you displayed the enhancement CO vs the enhancement $CO_2$ (e.g. the background subtracted off each parameter).

Here we like to refer to our reply to the main comment.

Pg 15 Ln 308: Given your stated measurement uncertainty, can you really say this is a significant change?

Since the difference of 0.8 per mill is larger than the measurement uncertainty mentioned in the methods section (0.5), it is statistically significant. Yet, Ln 308 tried to state that it is not sure if this is also larger than the natural variability. This is now restated.

**"Though the observed difference in $\delta^{13}C$ between the plume and the background sample is significant (0.8 ‰), but could also be the result of natural variability."**

Page 15 Table 4: Include data from 4-Sept if possible

As it is not used, see earlier replies concerning LISA measurements from 4 September; it is not added.

Pg 19, ln 376: Again, you are not presenting the enhancement ratios here. This is just the CO:$CO_2$ ratio from your overall flight. To get the enhancement ratio, you would subtract your background profile from your plume profile to plot the enhancement CO:$CO_2$ ratios.

Here we also like to refer back to the replay to the main comment.

**General comments Referee #2**

The paper "Wildfire smoke in the lower stratosphere identified by in situ CO observations" presents measurements of CO mole fractions, $CO_2$ mole fractions, and isotopic composition of CO from two balloon-borne instruments. Overall, the paper presents new datasets and includes in depth analysis of the measurements. It is generally well written, with clear discussion of results. The

paper would make a good contribution to ACP, after the following comments are addressed.

For the most-part, I found the description and justification for the methods chosen were complete and well-referenced. However, I do not have the expertise to fully evaluate some details of the data collection and analysis methods (Sect. 2.1, 2.2 2.3.2, 2.3.3). In some sections, I found it a bit difficult to follow the methods. For example, some descriptions of the methodology are written into figure and table captions. Throughout the paper, it would be helpful if high level information was provided at the beginning of each section to help guide the reader and explain how and why certain methods were applied. I have included some specific comments below, which include examples where additional descriptions would help.

We thank Anonymous Referee 2 for the kind words of appreciation for our manuscript. We hope to have addressed the issue of readability of the methods section by incorporating the essential information into the main text. Specifically, information that was only included in the captions of figures and tables.

**Specific comments from Referee #2**

Line 15: "The in situ observations provide information ... of the 2017 smoke plume" The closing sentence of the abstract is a bit unclear. Is the "information on the trace gas chemistry" what was described in the previous part of the abstract? If so, perhaps could use more specific language (e.g., what new information is provided by this study?). Also, I don't see much mention of the 1 km width of the plume in the text (I assume this inferred from Sect. 3.1?). If this is a key result, then maybe it warrants more discussion in text.

We have changed the closing statement of the abstract, to remove ambiguity. We have added explicitly to section 3.1 that the vertical extent of the observed plume is 1 km. Something that in the earlier version was left for the reader to determine from line 250 of the initial manuscript, as pointed out by the referee. We have repeated this observation in the conclusions.

Revised in the abstract:

**"Finally, the plume was extending over 1 km in altitude, as inferred from the observations."**

Added in the results section:

**"The plume, extending over roughly 1 km in altitude, was well above the tropopause . . . "**

Added in the conclusions:

**"present on two consecutive days, and extending over 1 km in altitude."**

Line 59 "Methods": A very short overview of what the AirCore vs LISA measure (mole fractions vs isotopic composition) and to what approximate vertical / temporal resolution would be helpful here.

This information has been included in the revised manuscript. See also the reply to the main comment 3 from referee 1 above.

Line 139: I found it very difficult to understand the method used for the back trajectories until I reached Fig 2 and the associated text in Sect. 3.2, which walks the reader through this process. Until reaching this figure, I found terms like "piece-wise manner" confusing. To remedy this, the authors could merge Sect. 2.3.1 directly into Sect. 3.2 or perhaps make Sect. 2.3.1 a bit more general with a forward reference to Sect. 3.2 (e.g., something like used back trajectories from CLaMS and a piecewise method illustrated in Sect. 3.2).

The second suggestion is adopted in the revised manuscript. Also, the text in Sect. 2.3.1 has been changed, to clarify the method used.

**"Therefore, a correction for the vertical displacement is determined in correspondence with CALIOP elastic backscatter-ratios at 532 nm (Winker et al., 2010) as illustrated in Section 3.2."**

Line 148: A bit of an overview would be helpful for Sect. 2.3.2 and 2.3.3. How are the methods being applied to the data? What information do the methods in Sect. 2.3.2 provide compared to Sect. 2.3.3? Have these methods been applied to similar datasets before?

We have added a statement to both Sect. 2.3.2 and 2.3.3 to clarify what information is retrieved from the data. We have added a few references where the reader can find examples of earlier application of the method specified Sect. 2.3.2 (added references: (Vimont, Turnbull, Petrenko, Place, Sweeney, et al., 2019; Vimont, Turnbull, Petrenko, Place, Karion, et al., 2017; Gromov and C. A. Brenninkmeijer, 2015)). The method in Sect. 2.3.3 is, to our best knowledge, not used in atmospheric science, and we therefore cannot provide a good example. The method in Sect. 2.3.3 is essentially an extension to the two end member case presented in Sect. 2.3.2. Sect. 2.3.3 has been revised, to improve clarity.

To section 2.3.3 (used to be 2.3.2) this sentence is added

**"The method usually employed to determine the source signature of an observed pollution, is to assume that a measured air parcel is the result of mixing of background air and a polluting source, i.e. two end-member mixing."**

To section 2.3.4 (used to be 2.3.4) this sentence is added

**"The mass balances in Equations 1 and 2 can be extended to allow**

**for more than two end members"**

Line 152: Did not define "f" in the text. Check throughout that all variables in equations are defined. In the revised manuscript $f$ is now defined. We have also taken a good look at other definitions of variables.

Line 191: I found this section confusing at first. I had to jump around the text to understand how Table 3 was constructed and how this information was used. Some of the methods/categories are only really described in the Table 3 and Figure 6 captions. Also, the Table 3 caption references "Monte-Carlo simulation" at the very top of the caption, but the MC simulation is not mentioned until much further in the text. One possible way to improve this would be to give a high level overview when the table is first introduced. This would describe why the table was put together, how the various lines of the table were compiled (with cross-references to relevant sections), the purpose / input parameters of the MC simulation when the table is first introduced.

Table 3 caption: The caption is a bit confusing –review/rewrite and maybe move some of the details into the text describing the methods. For example, "...does not significantly affect the results" – the results of what? The MC simulation?

Table 3: Where did the numbers for AP OH-Corrected come from? Are these related to the numbers given around line 337 – if so, how exactly were they chosen (they don't seem to exactly match any of the numbers given in the text)?

A paragraph is added to clarify the purpose of Table 3 and the origin of the variables displayed in Table 3. The authors agree, upon rereading the caption of Table 3 that it was unclear, and thus the caption has been revised. This information has also been put in the main text. Given the size of the revision, the authors refer to the revised manuscript, Section 2.3.4.

Line 237: "This thus differs ... in Table 2". Could you clarify this statement? It looks like the values in Table 3 fall within the range of Table 2 – but are a narrower part of the range (because from specific type of biomass burning?)

The statement has been changed. It is indeed a narrower part of the range specified in Table 2.

**"This is thus a subset from the range of signatures displayed in Table 1, which provide a more general summary"**

Line 240: Please add a bit of a description of the Monte-Carlo simulation. It's not clear what data is being used, what equation it's applied to, and what the desired output is until the reader gets to the caption of Fig 6.

We have added a paragraph to include this information in the methods section, see the revised manuscript 2.3.4.

Line 255: "The observed $CO_2$ . . . Which allowed for determination of enhancement ratio" Is the observed $CO_2$ difference significant for both LISA and AC? It looks like LISA is biased high compared with AC (e.g., non-enhanced day for LISA is comparable to biggest enhancement for AC). Also, why does the slight increase in $CO_2$ allow for enhancement ratio calculations?

Unfortunately, LISA samples appear to suffer from a bias in $CO_2$, see the LISA and AirCore comparisons in (Hooghiem et al., 2018). This is now stated in text Sect. 2.2.1. LISA $CO_2$ data was not used in the analysis presented by this manuscript. It was initially displayed for completeness of the data description. On a second thought, $CH_4$ was also measured by both LISA and AirCore. Yet this data has not been used nor presented in this manuscript. Therefore the $CO_2$ measurements of LISA have been removed from Fig. 1b and removed from Table 4, as they don't play a role in further analysis.

Line 259: This section could use an opening sentence describing what the back trajectories are being used for.

This section has been updated according to earlier comments.

Line 260: What is the "match distance"? Is this the minimum distance from the back trajectory to the CALIOP scan? Also, since the centre/upper altitude are not shown – is this the lower altitude? Is this starting from 13.3 km (based on Sect. 2.3)?

The Match distance is the horizontal distance of the trajectory location to the orbit track at the time of the satellite overpass. The distance of 250 km was chosen to balance between a coverage of the plume and a sufficient number of matches. It should be well below the geographical extent of the wildfire plume in order to follow the plume. The trajectory shown, was the trajectory with the lowest match distance. This was the lower trajectory of the 4 September observation, which is stated in the manuscript.

**"Wherever the distance was smaller than 250 km"**

And to the caption of Figure 2:

**"A match on 3 September between CALIOP and the lower CLaMS back-trajectory starting on 4 September"**

Figure 2 caption: ". . . the correspondence was sufficiently good". Does this mean the altitude correspondence between the back trajectory and CALIOP aerosol enhancement?

It means that the computed back-trajectory location was coinciding with aerosol enhancements. We have changed the text to be more exact. In the manuscript only locations where the CLaMS trajectory location was not at an observed aerosol cloud are shown. A line has been added into the main text to clarify this.

**"Other matches between back-trajectory location and CALIOP overpass was always at a layer of aerosol enhancement and no new back-trajectories were initialized. These match results are not shown."**

Line 277: How are the slope / uncertainty calculated?

The slope was computed using linear least-squares regression. The uncertainty is the standard error for the gradient. For background, the routine used for the computation is provided by python scipy package (scipy.stats.linregress). We have updated the manuscript using the more common 95 % confidence interval for the slope, which can be computed from the standard error of the gradient.

Line 307: How do you know that the plume is clearly not stratospheric? Or is it just clear that the plume is different from background?

The text has been revised, and now states that the plume is an enhancement compared to the background and has unusual high mole fractions of carbon monoxide:

**"clearly shows that the CO plume is different than the background"**

Line 332: How did you choose the ranges of possible values from Fig 5?

The text clarifies where these values came from. They were defined as the range provided by the orange curve in figure 5, i.e. the computation using the value of OH deemed the most representative for the plume.

Figure 5: The numbers in the legend aren't showing up correctly. (No value for the blue marker, reading at 1ˆ6 instead of 10ˆ6 for the orange and green markers)

A pdf was uploaded to ACPD with correct Figures, and we have not checked the document thoroughly enough after conversion. The correct figure can be found in the marked up version as well as the revised version.

Line 343: Why is it within the range for "biomass burning" in Table 2 but not within the range for "wildfire smoke" in Table 3?

Table 3 provides a narrower range, and is a subset of the range in Table 2. Table 2 provides a summary of our knowledge of isotopic source signatures. The values in Table 3 are a subset of Table 2 because of two criteria: 1) smoke from the fire comes from a boreal forest containing mainly C3 plants 2) the fact that the smoke rose to a very high altitude (above the tropopause), shows that the combustion temperature was very high, resulting in higher $\delta^{18}O$ values; see the discussion by Kato et al., 1999. This short discussion has been added to our methods section. What is excluded in this analysis is mixing in the troposphere. The OH chemistry is much less important in the 5 hour up-draft. Qualitative this would shift the isotopes more to range in Table 3. Mixing wildfire smoke

with depleted tropospheric air leads to lower $\delta^{18}$O and $\delta^{13}$C values; undoing the mixing would increase them. Revised paragraph:

**"In addition, the wildfire smoke signature, see Table 1, is subjected to a large variability due to the type of burned plants (categorised as C3 or C4, with a difference in photosynthesis), the burning temperature, and possibly the groundwater isotopic composition (Kato et al., 1999). Since the wildfire originated in Canada, the fuel consisted mainly of C3 plants which are typically more depleted in $\delta^{13}$C. Furthermore, the fire was energetic enough to trigger a pyro-Cb event, which makes it reasonable to assume that it was in an efficient burning regime, which typically leads to higher $\delta^{18}$O. Therefore, the range of isotopic composition of CO of the wildfire smoke assumed here is according to atmospheric measurements around forest fires (C. Brenninkmeijer et al., 1999), see Table 3. This is thus a subset from the range of signatures displayed in Table 1, which provide a more general summary."**

Line 414: You mention that "little is known of the CO isotopic composition". Do your measurements of the background air mass contribute to this knowledge? If so may be worth mentioning the background measurements in the abstract.

We have added the measurement to the abstract as per the suggestion.

**"In addition to CO mole fractions, CO$_2$ mole fractions as well as isotopic composition of CO ($\delta^{13}$C and $\delta^{18}$O) have been measured in air samples, from both the wildfire plume and background, ..."**

Line 426: "Yet another event was modelled ..." is this another wildfire event? If so, specify.

Yes, indeed another wildfire event was modelled. We have updated the text with more details about this event.

**"Cammas et al., 2009 modelled their observations of wildfire smoke from several fires in Canada and Alaska and they estimated the amount of polluted boundary layer air above the tropopause to be 15–20 %."**

**Technical corrections from Referee #2**

All technical corrections and suggestions below have been incorporated into the revised manuscript.

Line 10: "Back-trajectory analysis, performed with ... Date of 12 August 2017" Very long sentence. Consider breaking it up.

**"Back-trajectory analysis was performed with the Chemical Lagrangian Model of the Stratosphere (CLaMS), tracing the smoke's origin to wildfires in British Columbia with an injection date of 12 August 2017. The trajectories are corrected for vertical displacement due to heating of the wildfire aerosols, by observations made by the Cloud-Aerosol Lidar with Orthogonal Polarization (CALIOP) instrument."**

Line 13: "Colombia" $\longrightarrow$ "Columbia" (also, line 426, 434)

Line 145: "Wherever the distance was below", replace with "Wherever the distance was smaller than"

Line 245: "its" $\longrightarrow$ "it is"

Figure 2 caption: Check capitalization after periods. Also, last sentence is run on "CALIOP, the time" $\longrightarrow$ "CALIOP. The time"

Line 307: ", see Table 4" $\longrightarrow$ "in Table 4"

Line 308: "stratospheric" $\longrightarrow$ "the stratosphere"

Line 309: "it cannot be excluded" $\longrightarrow$ "it cannot be ruled out"

Line 356: delete repetition? "... was used to determine fractions of tropospheric and stratospheric air in the plume"

Line 414: "pollution air" $\longrightarrow$ "polluted air"

**Bibliography**

Brenninkmeijer, C.A.M et al. (1999). "Review of progress in isotope studies of atmospheric carbon monoxide". In: *Chemosphere - Global Change Science* 1.1-3, pp. 33–52. DOI: 10.1016/S1465-9972(99)00018-5.

Cammas, J.-P. et al. (2009). "Injection in the lower stratosphere of biomass fire emissions followed by long-range transport: a MOZAIC case study". In: *Atmospheric Chemistry and Physics* 9.15, pp. 5829–5846. DOI: 10.5194/acp-9-5829-2009.

Cohen, E.R. et al. (2007). *Quantities, Units, and Symbols in Physical Chemistry.* RSC Publishing. DOI: 10.1039/9781847557889.

Crosson, E.R. R (2008). "A cavity ring-down analyzer for measuring atmospheric levels of methane, carbon dioxide, and water vapor". In: *Applied Physics B* 92.3, pp. 403–408. DOI: 10.1007/s00340-008-3135-y.

Gromov, S. and Carl A.M. Brenninkmeijer (2015). "An estimation of the $^{18}O/^{16}O$ ratio of UT/LMS ozone based on artefact CO in air sampled during CARIBIC flights". In: *Atmospheric Chemistry and Physics* 15.4, pp. 1901–1912. DOI: 10.5194/acp-15-1901-2015.

Hooghiem, Joram J. D. et al. (2018). "LISA: a lightweight stratospheric air sampler". In: *Atmospheric Measurement Techniques* 11.12, pp. 6785–6801. DOI: 10.5194/amt-11-6785-2018.

Jost, Hans-Jürg et al. (2004). "In-situ observations of mid-latitude forest fire plumes deep in the stratosphere". In: *Geophysical Research Letters* 31.11. DOI: 10.1029/2003GL019253.

Kato, S. et al. (1999). "Stable isotopic compositions of carbon monoxide from biomass burning experiments". In: *Atmospheric Environment* 33.27, pp. 4357–4362. DOI: 10.1016/S1352-2310(99)00243-5.

Khaykin, S. M. et al. (2018). "Stratospheric Smoke With Unprecedentedly High Backscatter Observed by Lidars Above Southern France". In: *Geophysical Research Letters* 45.3, pp. 1639–1646. DOI: 10.1002/2017GL076763.

Mak, John E. et al. (2003). "The seasonally varying isotopic composition of the sources of carbon monoxide at Barbados, West Indies". In: *Journal of Geophysical Research: Atmospheres* 108.D20. DOI: 10.1029/2003JD003419.

Mauzerall, Denise L. et al. (1998). "Photochemistry in biomass burning plumes and implications for tropospheric ozone over the tropical South Atlantic". In: *Journal of Geophysical Research: Atmospheres* 103.D7, pp. 8401–8423. DOI: 10.1029/97JD02612.

Novelli, Paul C., L. Paul Steele, and Pieter P. Tans (1992). "Mixing ratios of carbon monoxide in the troposphere". In: *Journal of Geophysical Research: Atmospheres* 97.D18, pp. 20731–20750. DOI: 10.1029/92JD02010.

Sturm, P. et al. (2004). "Permeation of atmospheric gases through polymer O-rings used in flasks for air sampling". In: *Journal of Geophysical Research: Atmospheres* 109.D4. DOI: 10.1029/2003JD004073.

Tans, Pieter P. (1980). "On calculating the transfer of carbon-13 in reservoir models of the carbon cycle". In: *Tellus* 32.5, pp. 464–469. DOI: 10.3402/tellusa.v32i5.10601.

Vimont, Isaac J., Jocelyn C. Turnbull, Vasilii V. Petrenko, Philip F. Place, Anna Karion, et al. (2017). "Carbon monoxide isotopic measurements in Indianapolis constrain urban source isotopic signatures and support mobile fossil fuel emissions as the dominant wintertime CO source". In: *Elem Sci Anth* 5.63, p. 63. DOI: 10.1525/elementa.136.

Vimont, Isaac J., Jocelyn C. Turnbull, Vasilii V. Petrenko, Philip F. Place, Colm Sweeney, et al. (2019). "An improved estimate for the $\delta^{13}C$ and $\delta^{18}O$ signatures of carbon monoxide produced from atmospheric oxidation of volatile organic compounds". In: *Atmospheric Chemistry and Physics* 19.13, pp. 8547–8562. DOI: 10.5194/acp-19-8547-2019.

Winker, David M. et al. (2010). "The CALIPSO Mission". In: *Bulletin of the American Meteorological Society* 91.9, pp. 1211–1230. DOI: 10.1175/2010BAMS3009.1.